# A process-based anatomy of Mediterranean cyclones: From baroclinic lows to tropical-like systems

Emmanouil Flaounas[1], Suzanne L. Gray[2], Franziska Teubler[3]

[1]Institute for Atmospheric and Climate Science, ETH Zurich, Zurich, Switzerland
[2]Department of Meteorology, University of Reading, Reading, UK
[3]Johannes Gutenberg-Universität Mainz, Mainz, Germany

*Correspondence to*: Emmanouil Flaounas (Emmanouil.flaounas@env.ethz.ch)

**Abstract.** In this study, we address the question of the atmospheric processes that turn Mediterranean cyclones into severe storms. Our approach applies on-line potential vorticity (PV) budget diagnostics and piecewise PV inversion to WRF model
simulations of the mature stage of 100 intense Mediterranean cyclones. We quantify the relative contributions of different processes to cyclone development and therefore deliver, for the first time, a comprehensive insight into the variety of cyclonic systems that develop in the Mediterranean from the perspective of cyclone dynamics.

In particular, we show that all 100 cyclones are systematically influenced by two main PV anomalies: a major anomaly in
the upper troposphere, related to the baroclinic forcing of cyclone development, and a minor anomaly in the lower troposphere, related to diabatic processes and momentum forcing of wind. Among the diabatic processes, latent heat is shown to act as the main PV source (reinforcing cyclones), being partly balanced by PV sinks of temperature diffusion and radiative cooling (weakening cyclones). Momentum forcing is shown to have an ambiguous feedback, able to reinforce and weaken cyclones while in certain cases playing an important role in cyclone development. Piecewise PV inversion shows
that most cyclones develop due to the combined effect of both baroclinic and diabatic forcing, i.e. due to both PV anomalies. However, the stronger the baroclinic forcing, the less a cyclone is found to develop due to diabatic processes. Several pairs of exemplary cases are used to illustrate the variety of contributions of atmospheric processes to the development of Mediterranean cyclones: (i) cases where both baroclinic and diabatic processes contribute to cyclone development; (ii) cases that mainly developed due to latent-heat release; (iii) cases developing in the wake of the Alps; and (iv) two unusual cases,
one where momentum forcing dominates cyclone development and the other presenting a dual surface pressure centre. Finally, we focus on ten medicane cases (i.e. tropical-like cyclones). In contrast to their tropical counterparts—but in accordance with most intense Mediterranean cyclones—most medicanes are shown to develop under the influence of both baroclinic and diabatic processes. In discussion of medicane driving processes, we highlight the need for a physical definition of these systems.

## 1 Introduction

The Mediterranean basin is among the most cyclogenetic regions in the world (Petterssen 1956; Alpert et al. 1990; Maheras et al. 2001; Neu et al. 2013). Systems developing in this region range from weak orographic lows to intense cyclones or even tropical-like cyclones with severe socio-economic impacts (Nissen et al. 2010; Raveh-Rubin and Wernli, 2015). However, the term *Mediterranean cyclones* is typically used in the scientific literature to define a geographical origin rather than a cyclone category with specific dynamical characteristics. Indeed, the state-of-the-art is lacking a systematic analysis of the dynamics of these systems. In this regard, this study focuses on the most intense Mediterranean cyclones and in particular on the processes that turn these systems into severe storms.

Concerning the most intense Mediterranean cyclones, there is a general consensus that Mediterranean cyclogenesis is provoked by intruding upper-tropospheric systems that trigger baroclinic instability (Fita et al. 2006; Kouroutzoglou et al. 2011; Claud et al. 2010; Flaounas et al. 2015). Such systems typically correspond to potential vorticity (PV) streamers, equivalent to southern deviations of the Polar Jet that are a direct result of wave breaking over the Atlantic Ocean (Raveh-Rubin and Flaounas, 2017). After cyclogenesis takes place, baroclinic forcing may further intensify surface cyclones (e.g. Prezerakos et al. 2006; Lagouvardos et al. 2007) favouring, in parallel, convection in cyclone centres due to forced large-scale ascent. Recent studies used lightning observations to show that about one third of the most intense Mediterranean cyclones are associated with deep convection close to their centre (Galanaki et al. 2016; Flaounas et al. 2017). Due to latent heat release, convection serves as a net source of PV in the lower-to-mid troposphere and thus reinforces cyclonic circulations. Consequently, both baroclinic instability and convection may act in synergy to intensify Mediterranean cyclones (Fita et al. 2006; Chaboureau et al. 2012; Miglietta et al. 2017). Nevertheless, the processes that play primary roles in the development and intensity of Mediterranean cyclones is still an open question.

Medicanes (portmanteau of the words *Mediterranean hurricanes*) are rare cyclonic systems (1-3 occurrences per year) and have been proposed in several past studies to share similar dynamics with tropical cyclones (e.g. Emanuel, 2005; Fita et al. 2007). This similarity is mainly attributed to the visual characteristics of medicanes, i.e. a cloudless "eye" in the centre of a spiral cloud coverage (Tous and Romero 2013), but also due to cyclone phase diagrams that showed axisymmetric warm core structures (Cavicchia et al. 2014). To our knowledge the identification of all known medicane cases has been based on subjective criteria and, in particular, on the observation of cloud coverage satellite images (Tous and Romero 2013; Nastos et al. 2017). However, the similarity of medicanes to their tropical counterparts is not clear from the perspective of atmospheric processes. Although deep convection and wind-induced surface heat exchanges have been shown to sustain the cyclonic circulation of several medicane cases (e.g. Miglietta et al. 2013; Dafis et al. 2018; Miglietta and Rotunno, 2019), an upper-tropospheric PV streamer or a cut-off is typically located close to medicanes (Tous and Romero, 2013; Nastos et al. 2017). This suggests a non-negligible contribution of baroclinic forcing to medicane development. Indeed, several known medicane

cases have been shown to develop due to the combined effect of both baroclinic and diabatic processes (Fita et al. 2006; Chaboureau et al. 2012; Carrio et al. 2017; Miglietta and Rotunno, 2019). Furthermore, a recent case study of a medicane showed that the axisymmetric warm core structure may be reached when there is a warm seclusion and a relatively weak cut-off system (Fita and Flaounas, 2018). This raises a question of the driving processes of medicanes and whether other Mediterranean cyclones, not diagnosed as medicanes, are diabatically driven by convection in their centre.

Baroclinic instability and convection are two of the most common processes that affect the dynamics of extratropical cyclones. However, Mediterranean cyclones are also affected by the complex geography of the region. For instance, the relatively small size of the Mediterranean Sea and the arid areas of North Africa have an indirect effect on cyclones by limiting their supply of water vapour (Flaounas et al. 2019). On the other hand, the high mountains that surround the Mediterranean basin have a direct effect on cyclones by interacting with the air flow and triggering PV streamers of topographic origin. Such streamers are usually referred to as PV banners (e.g. Aebischer and Schär,1998) and correspond to filaments of diabatically produced PV at the wake of mountains (Rotunno et al. 1999; Epifanio and Durran, 2002; Schar et al. 2003; Flamant et al. 2004). The potential role of such PV production in Alpine cyclogenesis, one of the most cyclogenetic areas of the Mediterranean basin, has long been suggested to be important (e.g. Bleck and Mattocks, 1984) and have been recently reviewed by Buzzi et al. (2020). Especially when combined with the upper-tropospheric PV streamer, PV banners may even reinforce cyclones (Tsidulko and Alpert, 2001; McTaggart-Cowan et al. 2010a, 2010b). Therefore, the uniqueness of the region adds a higher degree of complexity to the processes implicated in Mediterranean cyclone development.

Cyclones may be conceptualized as the outcome of PV anomalies that extend to different atmospheric levels. t al.As a diagnostic variable, PV has been especially helpful to provide deep insights into the development mechanisms of different types of cyclones, including tropical cyclones (e.g. Möller and Montgomery, 2000; Chan et al. 2002) and mid-latitude storms (e.g. Boettcher and Wernli, 2011). As a result, the quantification of the relative contribution of different atmospheric processes to the total atmospheric PV is equivalent to delineating the contribution of these processes to cyclone dynamics. For instance, Gray (2006), Chagnon and Gray (2015) and more recently Spreitzer et al. (2019) used PV budget diagnostics to analyse the impact of diabatic processes to the dynamical structure of the upper troposphere. In the same direction, several studies used PV-budget diagnostics to investigate the role of convection and more generally of latent heat in the dynamical structure of cyclones (Martínez-Alvarado et al. 2016; Büeler and Pfahl, 2017; Attinger et al. 2019). By extension, piecewise PV inversion of these individual contributions is expected to reproduce the partition of the low-level wind circulation that is related to each atmospheric process, i.e. the contribution of these processes to the cyclone itself (defined as a vortex).

Applying piecewise PV inversion to individual PV tracers is a well-established approach to quantify the relative contribution of different processes to the development of cyclones (e.g. Davis and Emanuel 1991; Wu and Emanuel 1995; Stoelinga, 1996; Huo et al. 1999; Bracegirdle and Gray 2009; Schlemmer et al. 2010; Seiler, 2019). For instance, Stoelinga (1996) and

Ahmadi-Givi et al. (2004) quantified the contribution of diabatic processes to the intensification of two mid-latitude storms. The use of PV-budget diagnostics or piecewise PV inversion to perform a process-based classification of cyclones is still missing from the state-of-the-art. Such an approach would extend the analysis of Čampa and Wernli (2012) who analysed the vertical PV profile of extratropical cyclones of different geographical origin and would be complementary to earlier efforts by Deveson et al. (2002), followed by Plant et al. (2003) and Gray and Dacre (2006) where extratropical cyclones were classified to three types according to the contribution of upper and lower troposphere to their development. Despite the considerable advancements in analysing and classifying tropical and mid-latitude cyclones from a PV perspective, Mediterranean cyclones have been rarely been the object of comprehensive studies in which their dynamical structure is analysed in a climatological context. In this study, we use regional climate modelling to simulate 100 of the most intense Mediterranean cyclones and we analyse their dynamics using PV-budget diagnostics and piecewise PV inversion. In particular, we address the following specific objectives:

1 - Quantify the relative contribution of different processes to the dynamics of Mediterranean cyclones, defined by the spatial variability of PV during their mature stage; and

2 - Delineate the role of different processes in cyclone development and thus provide insights into the yet uncharted theoretical continuum between tropical-like cyclones (i.e. "purely" diabatic) and baroclinic lows.

In the next section, we introduce the modelling, diagnostics and PV inversion methods that we used to address these objectives. Section 3 places the simulated cyclones in a climatological context and then sections 4 and 5 present our main results. Finally, section 6 hosts a summary of the analysis and the conclusions of this study.

## 2 Methods

### 2.1 Cyclone cases selection and modelling approach

To build a climatology of cyclones, we applied the cyclone tracking method of Flaounas et al. (2014) to the hourly relative vorticity fields at 850 hPa, taken from the ERA5 reanalysis of the European Centre of Medium-Range Forecasts in the 10-year period from 2008 to 2017 (Copernicus Climate Change Service; C3S, 2017). Out of all tracked cyclones, we retained the 100 most intense ones. However, the tracking method is sensitive to identifying long lasting relative vorticity local maxima and in several cases these maxima lacked a clear cyclonic structure, i.e. they did not correspond to structured wind vortices. These tracked features were characterised as "false positives" and were subjectively excluded from our dataset of 100 most intense cyclones. Our main criterion to retain a system is the existence of a clear relative vorticity maximum that is outlined by enclosing mean sea level pressure contours. The relative vorticity and mean sea level pressure fields at the time of cyclones' mature stages are shown in the supplementary material. Out of 100 cyclones, ten cases were diagnosed as

medicanes in recent studies (Nastos et al. 2018; Di Muzio et al. 2019). These cases are shown as cases 7, 14, 24, 28, 46, 56, 65, 69, 75, 76 of the supplementary material.


A simulation has been performed for each individual cyclone, using the Weather, Research and Forecasting model (WRF, version 4.0; Skamarock et al. 2008). All simulations have been initialised 36 hours before the cyclone's mature stage, at the time of maximum relative vorticity in ERA5, and last for 72 hours in total. Initial and boundary conditions have been taken from the hourly fields of the ERA5 reanalysis at 0.25 degrees of grid spacing. All the simulations used a common domain

(shown in Fig. 1) which is composed of 350x200 grid points with a grid spacing of 15 km. In the vertical, the model top was set at 50 hPa and we used 34 hybrid eta levels that follow terrain topography in the lower troposphere, gradually transitioning to constant pressure levels in the higher troposphere. The model parametrisation was also common for all simulations and consists of the Single-Moment five-class microphysics scheme (WSM5; Hong et al. 1998, 2004), the Dudhia shortwave radiation scheme (Dudhia 1989), the Rapid Radiative Transfer Model longwave radiation scheme (Mlawer et al.

1997), the Yonsei University planetary boundary layer (PBL) scheme (Hong et al. 2006; Hong, 2010), and the Kain-Fritsch convective parameterization (Kain 2004). This combination of physical parametrisation was shown to adequately reproduce Mediterranean cyclones (e.g. Flaounas et al. 2019; Fita and Flaounas, 2018; Miglietta and Rotunno, 2019). Especially the use of a five-class microphysics scheme was shown to yield simulations with a small track error in two medicane cases, when compared to the use of other microphysical parametrisations (Miglietta et al, 2015; Pytharoulis et al. 2018). It is

however noteworthy that the absence of hail or graupel results in unrealistic representation of the microphysical properties of deep convective clouds.

**2.2 Potential vorticity tracers**

A module has been implemented into WRF that calculates PV at every model time-step according to the following equations:


$$Q = \frac{-g}{\mu} \xi_\alpha \cdot \nabla \theta \text{ (Eq. 1),}$$

$$\frac{dQ}{dt} = \frac{-g}{\mu} \left( \xi_a \cdot \nabla \dot\theta + \nabla \times F \cdot \nabla \theta \right) \text{ (Eq. 2),}$$

where $Q$ is PV, $g$ is the acceleration due to gravity, $\mu$ is the atmospheric column dry air mass, $\boldsymbol{\xi_\alpha}$ is the three dimensional

field of absolute vorticity, $\boldsymbol{F}$ is a two dimesnional field that represents a mixing wind acceleration term that includes surface friction, turbulent and diffusive processes, $\theta$ is potential temperature and $\dot\theta$ is its time derivative. PV is calculated on eta ($\eta$) model levels that range from 0 to 1 ($\eta = (P_\eta - P_s)/\mu$, with $P_\eta$ representing pressure on eta level $\eta$ and $P_s$ the surface pressure).

Equations 1 and 2 follow the formulation of Cao and Xu (2011) that has the advantage of avoiding wind and temperature interpolations—or approximations—in atmospheric levels other than the ones in model outputs. After calculating total PV, a

PV-budget diagnostic is activated at every model time step. This diagnostic follows previous methodological approaches where the total atmospheric PV is decomposed into several non-conserved partitions and one conserved partition (Lamarque and Hess, 1994; Stoelinga, 1996; Gray 2006). All these PV partitions are treated by the model as scalars, subject to advection and thus hereafter are referred to as PV tracers.

Each non-conserved PV tracer is attributed to one parametrised process that acts as a source or a sink of atmospheric PV. Overall, six non-conserved PV tracers are used. These derive from latent heat release by microphysics ($q_{mp}$) and convection ($q_{cu}$) parametrisations; turbulent fluxes of temperature by the planetary boundary layer parametrisation ($q_{bl}$); atmospheric warming/cooling by shortwave ($q_{sw}$) and longwave ($q_{lw}$) radiation; and a mixed term of momentum acceleration applied to air masses ($q_{mo}$), which is also provided by the boundary layer parametrisation. As explained below, the conserved PV tracer

($q_{co}$) represents the contribution of the large-scale flow to PV and is only subject to advection. In contrast, non-conserved PV tracers are, in addition to advection, also subject to accumulations of net gains and losses of PV. These gains and losses of PV represent the right-hand side terms of Eq. 2 and derive from the net temperature and momentum forcings of the model physical parameterisations to the primitive equations. More precisely and following the annotation in Stoelinga (1996), each PV tracer is calculated at every model grid point as follows:


$$q_x(\tau) = q_x(0) + \int_1^\tau \left( -v \cdot \nabla q_x + G_x \right) dt \text{ (Eq. 3)},$$

$$\text{where} \quad \begin{aligned} G_x &= \frac{-g}{\mu} \xi_a \cdot \nabla T_x \\ G_{(x=mo)} &= \frac{-g}{\mu} (\nabla \times F) \cdot \nabla \theta \end{aligned} \quad \text{(Eq. 4).}$$

$v$ represents the three-dimensional wind field and $q_x$ represents any PV tracer at a given time step $\tau$, related to process $x$ (co, mp, cu, bl, sw, lw or mo). The term $q_x(0)$ represents the value of PV tracers in the model's initial conditions, while PV sources and sinks are represented by the term $G_x$. The latter is calculated by Eq. 4, using the full model wind outputs and the temperature tendencies from model parametrisations (represented by the term $T_x$). Finally, it is noteworthy that $G_{mo}$ is calculated using $F$ from the boundary layer parametrisation and the full model $\theta$.


Given that $q_{co}$ represents the PV partition that is related to the large-scale flow, the total atmospheric PV at the initial time ($q_{co}(0)$ in Eq. 3) and boundary conditions are entirely attributed to $q_{co}$, while the term $G_x$ is set to zero in Eq. 3. As a result, $q_{co}$

is constantly introduced into the domain by prescribing the full atmospheric PV of the large-scale flow—regardless of whether it is produced by non-conservative processes—and it is interpreted as the PV partition that incorporates the dynamics that are "external" to the simulation domain (i.e. the Mediterranean region). On the other hand, non-conserved PV tracers at initial time ($q_x(0)$ in Eq. 3) and boundary conditions are all set to zero. This approach suggests that diabatic processes start taking place only after the initial conditions of the simulation and only within the simulation domain. Setting the initial conditions 36 hours before the cyclone's mature stage is a rather sufficient period to capture cyclogenesis. Indeed, the characteristic lifetime of Mediterranean cyclones is of the order of two days (Flaounas et al. 2015). Therefore, non-conserved PV tracers accumulate PV sources and sinks due to specific processes ($G_x$ term in Eq. 4) and thus "memorize" the processes that lead cyclones to intensify. At any given model time step, the total PV equals the sum of all tracers:

$$Q = q_{co} + q_{mp} + q_{cu} + q_{bl} + q_{sw} + q_{lw} + q_{mo} + q_r \text{ (Eq. 5).}$$

As for PV tracers, potential temperature may be also treated as a tracer ($\theta$ tracers hereafter), subject to advection and to accumulations of temperature tendencies ($T_x$) as produced from the physical parametrisations. Consequently, one conserved and several non-conserved $\theta$ tracers may be considered, similarly to PV (except for momentum as that does not apply direct changes to temperature):

$$\theta = \theta_{co} + \theta_{mp} + \theta_{cu} + \theta_{bl} + \theta_{sw} + \theta_{lw} + \theta_r \text{ (Eq. 6).}$$

Models are not numerically bound to conserve PV at every model time step (Saffin et al, 2016). Therefore, in Eqs 5 and 6, a residual occurs represented by the term with the subscript $r$. This residual can be mainly attributed to the dynamical core of the model and to adjustments made by the model to the prognostic variables in the primitive equations. However, the residual term does not attain large values and its average within a 300 km radius around cyclone centres is of the order of 0.2 PVU. Higher residuals are only found in the first model level with a median of -0.17 PVU, but ranging between -0.5 and -1 PVU in 14 cases out of 100. Finally, it is noteworthy that version 4.0 of the WRF model uses moist potential temperature to allow consistent treatment of moisture in the calculation of pressure in its dynamical core. For the purposes of our analysis we could also use the moist potential temperature as a basis for calculating a generalized PV field for a moist atmosphere. Schubert et al. (2001) showed that such a moist PV field still complies with the invertibility principle and thus it incorporates all information that defines the atmospheric state. However, to be consistent with previous studies, we performed our analysis with all PV fields in WRF being calculated with potential temperature.

**2.3 Piecewise PV inversion**

The contribution of different processes to cyclonic wind circulation has been determined by inverting individual PV anomalies under nonlinear balance (Charney 1955; Davis and Emanuel, 1991). Following Stoelinga (1996), every PV tracer may be considered as a PV anomaly, except for $q_{co}$, which is approximately equal to the full PV field in the upper troposphere. Therefore, $q_{co}$ anomalies have been defined as differences from three-day averages of $q_{co}$, i.e. the duration of each simulation. A three-days time period might be short to define PV anomalies in cases of slowly evolving upper

tropospheric systems. However, anomalies need to be consistently defined for all 100 cyclones and Mediterranean cyclones are systems of relatively short lifespan. Therefore, we consider that a three-days average is a fair compromise for the application of piecewise PV inversion in this study. To perform piecewise PV inversion we used the "subtraction approach" of Davis (1992) which has already been applied in past studies to separate the advective contributions of upper-tropospheric and low-tropospheric PV and potential temperature anomalies to upper-tropospheric PV anomalies (Piaget et al. 2015;

Teubler and Riemer 2016; Schneidereit et al. 2017 and Teubler and Riemer 2020). According to this approach, the partition of the atmospheric state (wind and geopotential) due to individual PV tracers (hereafter referred to as the balanced fields) is obtained by subtracting the balanced state related to the full PV inversion from the state related to the full PV minus the individual PV tracers. For instance, the balanced wind field, $\bar{U}$, due to $q_{mp}$ is equal to

$$\bar{U}\left(q_{mp}\right) = \bar{U}\left(Q\right) - \bar{U}\left(Q - q_{mp}\right) \text{(Eq. 7).}$$

All model outputs were interpolated to a regular longitude-latitude grid with a spacing of 1 degree and to pressure levels ranging from 950 to 100 hPa with a step of 50 hPa. Finally, following Stoelinga (1996), the piecewise PV inversion of individual PV tracers was performed by prescribing the relevant $\theta$ tracers in the boundary conditions at the top and bottom

atmospheric pressure levels as Neumann boundary conditions (925 and 125 hPa respectively), except for $q_{mo}$ which is not associated with a relevant $\theta$ tracer.

Numerous sensitivity tests have been performed to converge on a universal application of piecewise PV inversion for all 100 simulations. The method results have first been evaluated on the basis of accurate cyclone locations, by comparing cyclone

locations in the balanced relative vorticity fields (calculated using the balanced wind outputs from PV inversion) to the ones in the model outputs. Despite the use of interpolated fields, the balanced wind fields showed remarkably accurate cyclone locations with an average bias of less than 1 degree. In addition, the validity of the method has been evaluated through the comparison in the cyclone centres of (1) the balanced vorticity field at 850 hPa after inverting the full PV with (2) the sum of the balanced vorticity fields at the same pressure level after inverting the conserved and all non-conserved PV tracers. After

applying PV inversion we have retained 94 cases. Results showed that for these cases, the ratio between the two fields ranged between 0.9 and 1.1. This suggests that the inversion processes of individual PV tracers may accurately explain the

contribution of the different processes to cyclonic circulation by at least 90%. In the other cases six cases, the ratio exceeded 1.1, presenting a maximum of 1.4.

### 3. The 100 cyclone cases in a climatological context

Figure 1 shows the locations and depths of the 100 simulated cyclones at the time of their mature stage (i.e. when they reach maximum relative vorticity at 850 hPa). The cases tend to be concentrated over maritime areas, especially over the western Mediterranean Sea. Fewer systems are found in the eastern Mediterranean, while a small group of systems reach their maximum intensity over the continental areas of North Africa. Concerning the seasonal distribution of their occurrence, Fig. 2a shows that most cyclones occur in winter with fewer in autumn and spring, while none have been observed in summer with the exception of three cases in June. Although there is a certain arbitrariness in the selection of the cases, the spatial and seasonal distribution of the 100 cases is consistent with previous climatological studies of intense Mediterranean cyclones (Campins et al. 2011; Flaounas et al. 2018).

Similarly, the mean sea level pressure in the centre of our 100 cyclones is comparable to relevant climatological distributions. Indeed, Fig. 2b shows that the 5th, 50th and 95th simulated cyclones present values of 985, 998 and 1005 hPa, respectively. This is comparable to the results of Flaounas et al. (2018) where intense cyclones were tracked in different regional climate models during a 20-year period. In their results the 5th, 50th and 95th percentiles of cyclones depth roughly ranged between 1000-1005 hPa, 995-1000 hPa and 985-990 hPa, respectively (their figure 8). Therefore, our cyclone cases may be plausibly considered as representative of the climatology of intense Mediterranean cyclones.

Finally, to gain an insight into the dynamics of the 100 cyclones, Fig. 3 shows the composite averages of equivalent potential temperature and wind at 850 hPa for all 100 simulated cyclones, as well as their mean sea level pressure and PV at 300 hPa. Composite fields are spatially centred on the cyclone centres and temporally centred at the time of cyclones' mature stage (i.e. when cyclones have their maximum relative vorticity). Figure 3 clearly shows that cyclones develop in a baroclinic environment where equivalent potential temperature depicts frontal structures. The composite PV streamer is also clear at 300 hPa, presenting a thinning north-south structure mostly overlapping with the composite cyclone's cold sector. The highest PV values are slightly to the south of the composite cyclone's centre. These results are in fair agreement with previous studies on the dynamical structure of 200 intense Mediterranean cyclones, tracked in a 20-year climatology (Flaounas et al. 2015; their figure 3),

## 4. A process-based anatomy of Mediterranean cyclone dynamics

In the following, we analyse the dynamical structure of the 100 cyclones using vertical and horizontal profiles of the full atmospheric PV and PV tracers. Vertical profiles have been averaged within a radius of 100 km around cyclone centres. This radius is small compared to the characteristic size of the cyclones, but sufficient to encompass PV anomalies in their centres. This analysis will provide insight into the processes that form PV anomalies close to cyclone centres and thus play an important role in their dynamics.

### 4.1 Dynamical structure of cyclones: conserved and non-conserved PV

- *Total PV vertical profile*: All cyclones present a similar vertical PV profile (Fig. 4a) with a local maximum of the order of 1.3 PVUs at 850 hPa and a dramatic increase of PV above 400 hPa. Campa and Wernli (2012) showed that such a profile is typical for extratropical cyclones where the local maximum of PV at lower levels is related to diabatic heating and high PV values in the upper troposphere are related to PV streamers or, more generally, stratospheric air masses that overlap with cyclone centres. Indeed, the PV values in Fig. 3 are consistent with the ones in the vertical profiles of Fig. 4.

- *Conserved PV*: The PV streamer in Fig. 3 corresponds to stratospheric air that intrudes into the Mediterranean and thus is prescribed in the boundary conditions of the domain. As a result, the conserved component of PV in levels above 400 hPa ($q_{co}$; Fig. 4b) presents values comparable to the total PV (Fig. 4a). Indeed, the fields of both $q_{co}$ and total atmospheric PV present similar amplitude and shape when comparing their horizontal cross sections at 300 hPa in Figs 5a and 3, respectively.

In contrast to the upper troposphere, the vertical profile of $q_{co}$ tends to be rather invariant below 400 hPa, presenting an average value of the order of 0.5 PVUs (Fig. 4b). However, $q_{co}$ at low levels exceeds 1 PVU in several cases, especially in the first model levels. Figure 5b shows that higher values of $q_{co}$ are related to a low-level PV streamer that overlaps with the cold sector of the cyclones and wraps cyclonically around their centre. Being only subject to advection, $q_{co}$ is strongly dependent on the total PV in the initial conditions of the simulations. The origin of this low-level streamer is plausibly related to prominent values of PV close to orographic features. Indeed, the initial conditions of PV at 850 hPa are consistent with Fig. 5b, showing values of the order of 0.5 PVUs close to mountains (not shown).The role of mountains in acting as PV sources has been demonstrated previously by Schär and Durran (1997). The authors used idealised simulations to show that PV anomalies may be generated in the wake of mountains due to internal dissipation of the air flow. Therefore, mountains plausibly act as a source of relatively high PV values that potentially reinforces cyclones. This hypothesis has been confirmed in a case study by McTaggart-Cowan et al. (2010b). The authors performed several sensitivity tests on the dynamics of a medicane that developed close to the Alps. Results showed that removal of the mountains had a detrimental effect on cyclogenesis, inhibiting its occurrence.

- *Non-conserved PV*: The non-conserved component of PV is represented by the sum of all PV tracers apart from $q_{co}$ (i.e. $q_{mp}+q_{cu}+q_{bl}+q_{sw}+q_{lw}+q_{mo}$). Figure 4c presents a vertical alternation of signs: negative values are observed from the surface (around 975 hPa) to 950 hPa, thereafter positive values exist from 950 to 600 hPa, with a maximum at 850 hPa, and finally, negative values are observed between roughly 600 and 300 hPa. From the point of view of atmospheric dynamics, this alternation of signs suggests that diabatic processes tend to act as a sink of PV close to the surface, as a source in the lower-to-mid troposphere, and again as a sink in the upper troposphere. It is noteworthy that $q_{sw}$ presents especially low values with an average in cyclone centres that is close to zero (not shown). Therefore, analysis solely based on $q_{sw}$ is omitted hereafter.

In the lowest layer (from surface to 950 hPa), the negative values of the non-conserved PV tracers tend to get balanced by the vertically invariant positive values of $q_{co}$ (Fig. 4b), resulting in a PV average close to zero (Fig. 4a). In the middle layer (from 950 to 600 hPa), the diabatic processes are strongly characterized by the local maximum at 850 hPa (Fig. 4a). This local maximum is expected to be further enhanced by the typically positive values of low-level $q_{co}$ (Fig. 5b). However, Fig. 5d shows that the low-level local maximum of non-conserved PV tracers tends to be concentrated close to cyclone centres. This characteristic is in contrast to the broader, low-level conserved PV-streamer in Fig. 5b, suggesting that diabatic processes are expected to be directly related to the low-level cyclonic circulation. Finally, in the upper troposphere (from 600 to 300 hPa), the non-conserved PV tracers act as a rather weak sink of PV. Indeed, Fig. 5c presents significantly weak values when compared to the ones shown in Fig. 5a. Nevertheless, non-conserved PV tracers tend to erode the amplitude of the upper-level PV streamer and thus partly contribute to the thinning of its meridional structure in Fig. 3. It is noteworthy that values in Figs 5a and 5c are rather smoothed since they correspond to composite averages. In single cases, the shape of PV streamers is expected to present sharp edges due to diabatic processes that may locally act as strong sinks of PV (e.g. Gray, 2006; Chagnon et al. 2013; Spreitzer et al. 2019).

## 4.2 Dynamical structure of cyclones: contribution of processes to non-conserved PV

- *Latent heat:* Figure 4d shows that latent heating acts as a major source of PV within the lower and mid troposphere. Indeed, positive values of $q_{mp}+q_{cu}$ are observed from surface to 500 hPa. On the other hand, $q_{mp}+q_{cu}$ presents rather weak negative values from 500 to 300 hPa, acting as a sink of PV. As deduced from Eq. 2, this alternation of sign in Fig. 4d is related to the vertical gradient of latent heating. Indeed, Fig. 6a shows that $\theta_{mp}+\theta_{cu}$ presents its maximum value at around 500 hPa, i.e. the level where its vertical gradient turns from positive to negative.

Therefore, the vertical profile of $\theta_{mp}+\theta_{cu}$ is fairly consistent with that of $q_{mp}+q_{cu}$. In contrast, the vertical profile of instantaneous tendencies of latent heat release ($T_{mp}+T_{cu}$) in Fig. 6b is not consistent, presenting its local maximum at 800 hPa, i.e. in lower levels than 500 hPa as is the case for $\theta_{mp}+\theta_{cu}$ in Fig. 6a. Given that $\theta_{mp}+\theta_{cu}$ accumulate and transport instantaneous tendencies of latent heat release, this inconsistency can be attributed either one of, or most probably to the

345 combination of, two possible effects: convection is weakened at the time of the cyclones' mature stage when compared to previous times and high $T_{mp}+T_{cu}$ values are upwardly advected within the convective atmospheric column. To clarify this issue, we used the tracks of all 100 cyclones and produced similar profiles to Fig. 6b at a time 12 hours before the cyclones' mature stage. Results showed that the local maximum of $T_{mp}+T_{cu}$ was indeed located at an atmospheric level close to 500 hPa (not shown), plausibly suggesting that this inconsistency is due to convection being weakened when cyclones reach their

mature stage. As a result, the $q_{mp}+q_{cu}$ tracers are capable of including the "dynamical memory" of convection from previous times, i.e. convection may still reinforce cyclones in subsequent times to its maximum activity. This result is consistent with Miglietta et al. (2013) and Galanaki et al. (2016) who found that the maximum intensity of Mediterranean cyclones is preceded by deep convection in cyclone centres, but also with Price et al. (2009) and Whittaker et al. (2015) who showed that the maximum intensity of tropical cyclones comes in subsequent times, after the peak of convective activity. It is also

noteworthy that $\theta_{mp}+\theta_{cu}$ presents negative values from the surface to 850 hPa. Recent results on the water budget of Mediterranean cyclones showed that these negative values are related to rain evaporation (Flaounas et al. 2019).

Regarding the vertical level of latent heat contribution to cyclone dynamics, Fig. 4d shows that $q_{mp}+q_{cu}$ presents its maximum at 900 hPa, i.e. below the local maximum of non-conserved PV at 850 hPa (Fig. 4c). Figure 7a shows that $q_{mp}+q_{cu}$ acts indeed

as a significant source of PV at 900 hPa. However, the positive values of $q_{mp}+q_{cu}$ are spread along a southwest-to-northeast direction, i.e. they follow the frontal structure of the cyclones where convection is favoured. Comparing this spread of positive values of $q_{mp}+q_{cu}$ to the narrowly concentrated positive values for all non-conserved PV tracers in Fig. 5d, it is rather evident that other PV tracers act as sinks of PV, tending to counterbalance the PV sources due to latent heat, especially along the cyclone fronts. It is noteworthy that although frontal structures are evident in Figs 3 and 7, their exact location, intensity

and vertical PV profile are case dependent. As a result, latent heating-produced PV maxima along the cold fronts and their associated PV tendencies right below and above these maxima (Lackmann 2002) will be at some extent inconsistent between the cyclones. This will affect values in the composite averages of Fig. 7a. Such inconsistencies are also expected in composite averages of other PV tracers close to the surface (e.g. Figs 7c and 7e), where PV is typically negative in the cold sector, due to small or negative static stability, and positive values in the warm sector (Vanniere et al. 2016).

Finally, despite the important role of sea surface temperature (SST) in modulating convection (e.g. Miglietta et al. 2011; Pytharoulis, 2018), in this study we found no seasonal dependence or direct relationship between SSTs and $q_{mp}+q_{cu}$ in low atmospheric levels (not shown). This is partly due to the diversity of the 100 cyclones' dynamics and partly due to the fact that all systems occur in different regions and different seasons. For instance, convection might differ between two systems

with similar underlying SSTs due to different large scale forced ascent. Similarly, two cyclones that occur in the same season, but in different areas are very likely to present considerable differences in their underlying SSTs and consequently to convection. In fact, the investigation of the role of SST in cyclone dynamics from the perspective of PV tracers is not a straightforward issue due to the trait of $q_{mp}+q_{cu}$ PV tracers to accumulate the diabatically produced PV. As a result, an air

mass that has an impact on cyclone dynamics might have experienced the PV gains due to latent heat in remote areas to cyclones centre.

- *Momentum exchanges with environment:* The vertical profile of $q_{mo}$ is highly variable among the 100 cyclone cases (Fig. 4e). At the pressure level of 850 hPa, there are 73 cases where $q_{mo}$ is positive and thus functions as a source of PV (with a maximum of the order of 2 PVU) and 27 cases where $q_{mo}$ is negative and thus functions as a sink (with a minimum of the order of -0.5 PVU). This is consistent with Stoelinga (1996) who analysed the role of $q_{mo}$ in the development of a case study of a cyclone development and showed that indeed this PV tracer may act as both a source and a sink of PV. To understand the ambiguous contribution of $q_{mo}$ to cyclones development, we expand Eq. 4 into the following three terms:

$$G_{mo} = \frac{-g}{\mu} \left[ \left( \frac{\partial F_y}{\partial x} - \frac{\partial F_x}{\partial y} \right) \cdot \frac{\partial \theta}{\partial \eta} + \left( \frac{\partial F_x}{\partial \eta} \right)\left( \frac{\partial \theta}{\partial y} \right) - \left( \frac{\partial F_y}{\partial \eta} \right)\left( \frac{\partial \theta}{\partial x} \right) \right] \text{(Eq. 8)}$$

Adamson et al. (2006) used numerical simulations to analyse in depth the role of $q_{mo}$ in the development of an idealized cyclone. In their analysis, the first term is responsible for the "Ekman generation" of PV and is proportional to the contribution of the PBL parametrisation to the cyclone' s relative vorticity. As such, this term typically corresponds to a PV sink due to wind deceleration by frictional forces, i.e. frictional forces impose an anticyclonic circulation. On the other hand, the second and third term are responsible for the "baroclinic" generation of PV and are expressed by the dot product between the horizontal component of the curl of *F* and the horizontal gradient of $\theta$. In fair accordance with Adamson et al. (2006), Fig. 7b shows that positive values of $q_{mo}$ tend to be concentrated on the northeast side and close to the centre of the cyclones and negative values tend to be concentrated on the southwest side. The baroclinic generation of PV (i.e. the two last terms in Eq. 8) were shown by Adamson et al. (2006) to correspond to a significant PV source, where positive anomalies were suggested to be progressively advected by warm conveyor belts upwards and eventually polewards and towards the west side of cyclones centres. Therefore, the function of $q_{mo}$ as a source or a sink of PV in cyclones' centre depends on the vertical component of the curl of *F*, on the magnitude of baroclinicty and its relative location to the horizontal component of the curl of *F*, but also to the advection of $q_{mo}$ in cyclone centre. However, the relatively small size of the Mediterranean basin, its sharp land-sea transitions and the short lifetime of cyclones may not always allow a clear formation of frontal structures. It is thus plausible to suggest that the contribution of $q_{mo}$ to cyclone development may vary significantly from case to case with a general tendency to reinforce cyclones, i.e. the average in Fig. 4e presents positive values at 850 hPa. An exemplary case study is used in section 5.4 to provide further details on the baroclinic generation of PV in $q_{mo}$.

- *Turbulent fluxes of temperature*: The vertical profile of $q_{bl}$ (Fig. 4f) shows that turbulent fluxes of temperature tend to act as a source of PV within the very first model levels and as a sink right above. Indeed, Fig. 7c shows that $q_{bl}$ at 1000 hPa presents overall positive values, mostly concentrated along the cold front and close to the composite cyclone centre. Clearly, the values of $q_{bl}$ in Fig. 7c are unrealistically high, compared to typical values of total PV in the troposphere. Such high values may be attributed to the sharp vertical gradient of temperature fluxes within the surface layer, i.e. within the two first

model levels. Indeed, Fig. 6b shows a positive vertical gradient in the average profile of temperature turbulent fluxes ($T_{bl}$) in the first model levels. However, it is noteworthy, that the vertical profile of $T_{bl}$ is subject to diurnal variabilities of the PBL height affecting thus its vertical gradient and the magnitude of $q_{bl}$. In addition, $q_{bl}$ presents a high variability in amplitude and sign over the cyclone centres in Fig. 4f. It acts as a source of PV in the first model levels for 74 cases and as a sink of PV for the remaining 26 cases. For these 26 cases, the composite average of $q_{bl}$ at 1000 hPa presents a structure consistent with that

shown in Fig. 7c (not shown). However, the cyclone centre in these cases is slightly displaced with respect to the area where $q_{bl}$ presents high values. Consequently, the vertical profiles of $q_{bl}$ in Fig. 4f are relatively sensitive to the locations of the cyclone centres. Consistent with the monotonic negative vertical gradient of $\theta_{bl}$ in Fig. 6a, $q_{bl}$ acts as a sink of PV in the atmospheric levels above the PBL (roughly set at 950 hPa). Finally, negative values of $q_{bl}$ at 850 hPa (Fig. 7d) tend to overlap with the area where $q_{mp}+q_{cu}$ presents high positive PV values in Fig. 7a. This result suggests that turbulent fluxes of

temperature act as a sink of PV and thus tend to counteract the role of latent heat in reinforcing cyclonic circulation.

    - *Longwave radiative forcing of temperature*: The average vertical profile of $q_{lw}$ (Fig. 4g) shows overall negative values within the whole atmospheric column, especially within the PBL. These negative values are consistent with negative vertical gradients observed for both $\theta_{lw}$ and $T_{lw}$ in Figs 6a and 6b, respectively. As for $q_{bl}$, the vertical profile of $q_{lw}$ in Fig. 4g presents

unrealistically low PV values within the first model levels, when compared to total PV values in the troposphere. This is due to a sharp decrease of $T_{lw}$ within the first model levels. Indeed, positive values of $T_{lw}$ in the first model level are attributed to high concentrations of water vapour that absorb longwave radiation, while negative values in the levels above are mostly related to cooling, due to thermal radiation of air masses. In fact, the average vertical profiles of $q_{bl}$ and $q_{lw}$ in Fig. 4 seem to be symmetric and thus tend to counter each other, especially within the very first model level. Indeed, the correlation

coefficient between the values of $q_{bl}$ and $q_{lw}$ at the first model level for all 100 cyclones is -0.96. Moreover, Figs 7c and 7e show similar spatial distributions for the fields of $q_{bl}$ and $q_{lw}$. It is plausible that turbulent fluxes of water vapour and temperature are subject to the same physical mechanism and thus provoke similar vertical temperature gradients of $T_{bl}$ and $T_{lw}$ close to the surface. As a result, $q_{bl}$ and $q_{lw}$ are of similar amplitudes. On the other hand, and despite being highly correlated, $q_{bl}$ and $q_{lw}$ are not of equal absolute values. Differences between the 100 cyclone centres at the first model level

present an average of -1.3 PVUs and a standard deviation of 1.8 PVU. This suggests a high variability between the two fields, which in turn is strongly dependent on several factors such as background water vapour concentration and whether cyclone centres are located above the maritime or continental areas. In the atmospheric levels above the PBL, $q_{lw}$ presents

negative values mostly concentrated in cyclone centres (Fig. 7f). The sum of $q_{bl}$ and $q_{lw}$ (Fig. 4h) shows that the two PV tracers act as a sink of PV in the lower troposphere and thus are expected to weaken the cyclone.


- *Synopsis:* Upper-tropospheric PV is mostly composed of the conserved PV tracer ($q_{co}$), while the local maximum of PV anomaly at 850 hPa is mainly due to non-conserved PV tracers (Fig. 4c). Figure 8 summarizes the average PV tracer profiles in the lower troposphere. Latent heating and momentum acceleration are expected to amplify the cyclone by acting as PV sources ($q_{cu}+q_{mp}$ and $q_{mo}$, shown by the blue and black lines, respectively), while turbulent fluxes and cooling due to
longwave radiation are expected to weaken the cyclone by acting as PV sinks ($q_{bl}+q_{lw}$, shown by a red line). The vertical profile of latent heat PV tracers ($q_{cu}+q_{mp}$) has its maximum at 925 hPa, where latent heat release ($\theta_{cu}+\theta_{mp}$ in Fig. 6a) sharply increases before it reaches its maximum value at around 500 hPa. On the other hand, the profile of $q_{bl}+q_{lw}$ has its minimum at the level of 950 hPa, i.e. roughly at the top of the PBL. It is of no surprise that the vertical profiles of $q_{cu}+q_{mp}$ and $q_{bl}+q_{lw}$ in Fig. 8 tend to present vertically symmetric structures, longwave radiation and turbulent fluxes are expected to increase
proportionally with the intensity of convection. This direct relationship is evident in the scatterplot of Fig. 9, where $q_{cu}+q_{mp}$ and $q_{bl}+q_{lw}$ are compared at the level of 850 hPa for all 100 cyclones and are shown to present a correlation coefficient of -0.87. However, by considering mean sea level pressure in cyclone centres as a measure of intensity (depicted by dot sizes in Fig. 9), it is rather evident that there is no systematic pattern that relates $q_{cu}+q_{mp}$ or $q_{bl}+q_{lw}$ to cyclone intensity. Therefore, the role of other processes in cyclone intensification is expected to be crucial, especially momentum mixing and baroclinic
instability (i.e. the role of $q_{co}$ to cyclonic circulation).

## 5. Relating conserved and non-conserved PV to cyclone intensity

In the previous section we used PV tracers to gain a deeper insight into the role of different atmospheric processes in forming the dynamical structure of cyclones. However, delineating the contribution of baroclinic and diabatic processes to cyclone development is not a trivial task. To this end, we use piecewise PV inversion to define the partition of the
atmospheric state related to conserved and non-conserved PV. Figure 10 presents in a scatterplot the values of the balanced relative vorticity in cyclone centres at 850 hPa, calculated from the balanced flow fields after inverting $q_{co}$ anomalies and the sum of all non-conserved PV tracers (i.e. $q_{mp}+q_{cu}+q_{bl}+q_{sw}+q_{lw}+q_{mo}$). Red circles in Fig. 10 depict the medicanes in our dataset. Values correspond to averages within 300 km from the cyclone centres at the time of their mature stage.

Figure 10 shows a negative linear trend (correlation coefficient -0.65) which suggests that the more a cyclonic circulation is due to baroclinic forcing ($q_{co}$), the less it is due to diabatic processes. In fact, the majority of cases—about 60 cyclones— were found to be related to higher values of relative vorticity due to baroclinic forcing ($q_{co}$) than to diabatic processes. Most systems have positive values of balanced relative vorticity due to both baroclinic forcing and diabatic processes; however, negative values of vorticity in Fig. 10 are not automatically be interpreted as cyclolytic contributions to cyclone

development. For instance, negative values of balanced relative vorticity in Fig. 10 may be caused by a sharp anticyclonic steering of weak winds, embedded within a broader cyclonic wind field. Therefore, piecewise PV inversion is used in this section to better understand which processes play a primary role in cyclone development, rather than to quantify precisely the contribution of different processes to cyclone wind circulation. In the following sections we present and discuss several exemplary cases of Mediterranean cyclones.

**5.1 Synergistic forcing from baroclinic and diabatic processes**

Figure 11 shows the balanced wind and geopotential fields for two cyclone cases during their mature stage. Both cases have positive contributions to development from both baroclinic and diabatic processes according to Fig. 10. The first cyclone is located in the central Mediterranean and corresponds to a rather deep cyclone with a minimum mean sea level pressure value of 988 hPa (within the top 10% of our cases; Fig. 2). The second cyclone (Fig. 11b) is less deep (994 hPa) and is
located to the north of Sicily. Comparing the balanced vorticity fields from the full PV inversion (Figs 11a and 11b) and the cyclone centres from the simulation outputs (cyclones #5 and #78 in supplementary material), it is clear that PV inversion yielded an accurate representation of both systems.

Figures 11c and 11d show that both cases are clearly associated with a PV streamer that intrudes into the Mediterranean from
the north, wrapping cyclonically around the cyclone centres. Figure 11c shows that the cyclone centre is close to the local minimum of geopotential, as imposed by $q_{co}$ anomalies at 850 hPa, while the centre of the second cyclone in Fig. 11d is embedded within a broader and rather weaker cyclonic circulation. Consistent with the imposed circulation by $q_{co}$, both cyclone centres are also collocated with the local minimum of the geopotential field imposed by the non-conserved PV tracers at 850 hPa (Figs 11e and 11f). As a result, baroclinic and diabatic forcings synergistically contribute to the
development of both cyclones. Such a scenario is expected to be typical for most intense cyclones, the majority of which are concentrated within the range of positive relative vorticity values in Fig 10. Finally, it is noteworthy that Fig. 11e shows a filament of high PV values over the Aegean Sea with a south-north direction, and another one that extends southwards from the cyclone centre. Both filaments seem to outline the frontal structure of the cyclone and are coincident with the high relative vorticity values in both Fig. 11a and the supplementary material (cyclone #5).

## 5.2 Diabatically driven cyclones

Figure 12 presents two cases where diabatic processes contribute strongly to cyclone development. Both cyclones are located within the central Mediterranean and present a clear low-pressure centre. Especially, the cyclone in Fig. 12a is among the deepest ones in our dataset (982 hPa). In contrast, the cyclone in Fig. 12b is one of the weakest ones with a central mean sea level pressure of 1010 hPa. In both cases, $q_{co}$ presents a weak PV anomaly in the upper troposphere with a rather irregular shape (Figs 12c and 12d). In the first case, this anomaly imposes a rather weak balanced wind field at 850 hPa (Fig. 12c), while in the second case the imposed cyclonic circulation is clearly displaced from the cyclone's centre (Fig. 12d).

In contrast to the conserved component of PV, the non-conserved PV in Figs 12e and 12f presents high PV values that are concentrated in the cyclone centres. The inversion of such high values results in balanced geopotential fields with strong gradients, both presenting local minima over the cyclone centres (Figs 12e and 12f). To further clarify the role of convection in these two cases, Fig. 13 shows the fields of $q_{mp}+q_{cu}+q_{bl}+q_{lw}$ at 850 hPa, as well as their imposed balanced geopotential and wind fields. In both cases, the PV tracers are fairly similar to Figs 12e and 12f, while the imposed balanced fields clearly show a cyclonic circulation around the cyclone centres. Differences between Fig. 13 and Figs 12e and 12f are majorly attributed to $q_{mo}$ ($q_{sw}$ is negligible); however, these differences are not changing the primary role of convection in forming the cyclonic circulation. Therefore, Figs 12 and 13 present two cases where baroclinic instability plays a secondary role to their development and thus they can be considered diabatically driven cyclones.

## 5.3 Mountain forcing of cyclones

Figure 14 shows two examples that correspond to Alpine cyclogenesis. The relative vorticity field at 850 hPa presents high values that overlap with the western side of the Alpine mountain chain (Fig. 1) in both the actual model output and the balanced fields in Figs 14a and 14b. Cyclogenesis in the cases of Fig. 14—as for the majority of Mediterranean cyclones—is related to an intruding PV streamer. As shown in Figs 14c and 14d, during the mature stage of the cyclones, the streamer is located over the islands of Corsica and Sardinia imposing a cyclonic circulation over the region. In the first case (Fig. 14a), the cyclonic circulation associated with the full PV distribution is centred over the Gulf of Genoa and extends all over the Mediterranean region. In contrast, the streamer in the second case is located on the east side of a low-level ridge (Fig. 14b) and therefore its imposed cyclonic circulation is displaced towards the east, over the Adriatic Sea (Fig. 14d). Despite the differences between the two cases, both PV streamers impose strong northerly winds that pass over the western side of the Alps. These winds are expected to increase the horizontal gradient of wind speed between the western and eastern side of the Alps and thus to reinforce relative vorticity.

In both cases, PV banners clearly stand out as filaments of high PV values that enter into the cyclone centre from the western flank of the Alps (Figs 14e and 14f). Piecewise inversion of non-conserved PV tracers shows that these filaments impose a

cyclonic circulation that is centred over the Alps and reinforces the cyclones (Figs 14e and 14f). To better understand the processes leading to these filaments, Fig. 15 further decomposes the non-conserved PV tracers at 850 hPa into $q_{mp}+q_{cu}+q_{lw}+q_{bl}$ and $q_{mo}$. In both cyclones the high PV values are clearly related to high values of $q_{mo}$, while $q_{mp}+q_{cu}+q_{lw}+q_{bl}$

seems to play a secondary role among the non-conserved PV tracers. It is, however, important to stress that these cyclones would be unlikely to form without the upper-tropospheric PV streamers. et al.In fact, baroclinic forcing and topographic PV production contribute both to lee cyclogenesis. However, disentangling their relative contributions is a rather challenging issue since both processes take place in parallel (Buzzi et al. 2020 and references therein). Nevertheless, PV tracers may provide further insights into this complex problem. As discussed in section 4.1, mountains are likely to play the role of a

"constant" PV source due to their interaction with the air flow. This constant PV source is imprinted into the simulations' initial conditions and corresponds to the low-level streamer of $q_{co}$ that wraps cyclonically around the cyclone centres in Fig. 5b. However, Fig. 15 shows that large part of high PV values close to mountains is related to $q_{mo}$ and thus to Ekman and baroclinic PV production (last two terms in Eq. 8). In these regards, PV production plays an important role in lee cyclogenesis. This is consistent with the sensitivity tests of McTaggart-Cowan et al. (2010b) who showed that the removal of

the Alps inhibited cyclogenesis of a medicane.

After the stage of cyclogenesis, Mediterranean cyclones are still influenced by mountains interactions with the air flow due to the complex geography of the region and its sharp land-sea transitions. For instance, an air mass might reinforce a cyclone when it reaches to its centre due to experiencing earlier PV production while interacting with topography. However, the time

period during which air masses sustain their high values of $q_{mo}$ is rather uncertain, i.e. the "dynamical memory" of $q_{mo}$, similar to that discussed for latent heating in section 4.2. Therefore, it is rather difficult to discern the areas where the influence of mountains is more important. Such analysis would benefit from a Langrangian approach (e.g. Attinger et al. 2019) where the evolution and origin of high values of $q_{mo}$ could be backtracked to mountains. However, this is currently out of the scope of this study and will be the subject of a following research study.

**5.4 Two peculiar cases of Mediterranean cyclones**

In this section, we present two rather peculiar cases to illustrate the variability of the implicated dynamics in cyclone development.

The first case is presented in Fig. 16a where the cyclone centre is nested within an elongated, meridional structure of high

values of relative vorticity at 850 hPa. This elongated structure is fairly consistent with the direct model outputs (cyclone #24 in the supplementary material). However, in the model outputs there is also a secondary local maximum of relative vorticity south of the actual centre. The balanced fields in Fig. 16a instead capture this secondary centre as a southward extension of high values of relative vorticity. Such dual-centred cyclones have been found in several locations in the extratropics (Hanley and Caballero, 2012). Figure 16c shows that the primary cyclone centre (depicted by the magenta dot) is collocated with the

local minimum of the imposed geopotential field by $q_{co}$. Therefore, this cyclone is clearly intensified by baroclinic instability. Interestingly, the geopotential field imposed by the non-conserved PV also presents a local minimum, but this minimum is coincident with the secondary centre of the cyclone located to the south of the primary centre. Consequently, a dual centre of a cyclone may be formed due to different processes and a cyclone could be characterized as a diabatically or baroclinically driven system depending on the choice of its representative centre. A case of a Mediterranean cyclone with a

dual centre has been recently shown by Carrio et al. (2020). In their case study analysis, the authors showed that a secondary cyclone centre was nested within a main, primary cyclone. The latter corresponded to baroclinic low pressure cyclone, related to a clear frontal structure, while the secondary cyclone centre was related to a local pressure minimum generated by convection.

The second cyclone case (Fig. 16b) corresponds to one of the few cyclones in our dataset that presents exceptionally high values of $q_{mo}$ at 850 hPa. This rather deep cyclone (990 hPa) is located to the southwest side of Sardinia and is related to a rather "irregular" PV streamer, when compared to the previous exemplary cases (Fig. 16d). The streamer presents an east-west direction along 37$^o$N, enters the region from the Atlantic Ocean and has lower values of PV than the other examples shown. The geopotential field imposed by $q_{co}$ at 850 hPa is characterized by an anticyclonic circulation centred over Italy

and a cyclonic circulation centred over North Africa. The cyclone centre in Fig. 16b is located between these two opposite circulations and, therefore, $q_{co}$ does not contribute significantly to the cyclonic circulation, at least during the cyclone mature stage. On the other hand, the sum of all non-conserved PV tracers in Fig. 16f shows rather high values that are concentrated around the cyclone centre. This leads to a strong geopotential gradient in Fig. 16f centred over the cyclone centre. The $q_{mo}$ term represents about 80% of these high positive values (not shown) and therefore momentum forcing has a leading role in

the development of the cyclone.

In our knowledge, this is the first study that shows a cyclone system where the mature stage is self-sustained by momentum forcing within the PBL. It is however noteworthy that the convection parametrisation that we use in this study lacks application of momentum adjustments to the wind field. It is thus possible that part of the high $q_{mo}$ values in Fig. 16f is

related to the response of PBL parametrisation to convection. To test this hypothesis we performed an additional simulation where the Kain-Fritsch convective parameterization (Kain 2004) was replaced by the one of Tiedke (Zhang and Wang, 2017). The latter provides momentum adjustments, introduced to the model as an additional momentum forcing to the model primitive equations. Therefore, we introduced an additional PV tracer to Eq. 5. This new PV tracer is calculated similarly to $q_{mo}$, whereas $\boldsymbol{F}$ in Eq. 4 represents the momentum adjustments by the Tiedke convection parametrisation. Results from this

new simulation are highly consistent with the original simulation and $q_{mo}$ still presents exceptionally high PV values. In fact, the new PV tracer is of the order of 0.2 PVU in absolute values close to the cyclone centre (not shown).

This case study that we present in Fig. 16f is of particular interest for gaining further insights into the ambiguous role of $q_{mo}$ in acting as both a source and sink of PV in cyclone centres. For this reason, we further decomposed the PV budget by introducing two additional PV tracers. These new PV tracers combine the three terms of Eq. 8: the first PV tracer accumulates the Ekman PV generation due to the first term of Eq. 8 (hereafter $q_{mo1}$) and the second PV tracer accumulates the baroclinic PV generation due to the second and third term of Eq. 8 (hereafter $q_{mo2}$). Figure 17a shows the potential temperature at 850 hPa, along with the fields of wind, $q_{mo}$ and $q_{mo1}$ at the time of the cyclone's mature stage. The cyclone has a clear frontal structure where high values of $q_{mo}$ are concentrated along the warm front. This configuration is fairly similar to the idealized simulation results of Adamson et al. (2006). In contrast to $q_{mo}$, $q_{mo1}$ is acting cyclolytically, by imposing negative PV values around the cyclone centre. Therefore, it is clear that Ekman PV generation is weakening the cyclonic circulation and thus the high PV values of $q_{mo}$ close to the cyclone centre are by default related to $q_{mo2}$. To better understand this PV source, Fig. 17b shows the vertical cross section of $\boldsymbol{F}_x$ along the green dashed line of Fig. 17a. In addition, Fig. 17b shows the instantaneous baroclinic PV generation, i.e. the sum of the second and third terms of Eq. 8. PV sources are located within the PBL, in the eastern side of the cyclone (black contours in Fig. 17b) where the vertical gradient of $\boldsymbol{F}_x$ is negative. Although baroclinic PV production by momentum forcing within the PBL is expected to be common in extratropical cyclones, the conditions that lead $q_{mo}$ to dominate dynamics of a cyclone are yet to be determined. This will be the object of a forthcoming study.

### 5.5 Medicanes from a PV-based perspective

Medicanes have long been proposed to share similar dynamics with tropical cyclones (e.g Emanuel, 2005; Fita et al. 2007). Nevertheless, a PV streamer or a cut-off system is commonly present close to medicanes and therefore baroclinic forcing is expected to contribute to the development of these systems (Tous and Romero 2013; Flaounas et al. 2015; Nastos et al. 2018). This raises the question of whether medicanes can be considered as "purely" diabatic systems, driven by latent heat release as in their tropical counterparts. The ten cases included in this study have been diagnosed as medicanes due to their spiral cloud coverage and cloudless "eye" (Nastos et al. 2018; Di Muzio et al. 2019). The application of piecewise PV inversion to the conserved and non-conserved PV components is a sufficient method to identify whether these medicanes indeed share similar dynamics with tropical cyclones, or at least are "mostly" driven by diabatic processes.

Figure 10 places most medicanes (depicted by red circles) within the range of positive contributions from both baroclinic and diabatic forcings. However, two medicanes have close to a zero contribution from $q_{co}$, including medicane Trixie (Di Muzio et al. 2019) which corresponds to the case presented in Fig. 12b. In addition, a single medicane case is shown by Fig. 10 to be an outlier with considerable contributions to its relative vorticity from both baroclinic and diabatic processes: 5.7 and $2.5 \times 10^{-5}$ s$^{-1}$, respectively. The other medicanes in Fig. 10 are similar to the mixed cases presented in section 5.1 and thus the involvement of baroclinic forcing through the PV streamer cannot be neglected. This finding agrees with previous studies

where factor separation techniques showed that the PV streamer plays a primary role in the development of medicanes, especially when interacting with diabatic processes (Fita et al. 2006; Carrio et al. 2017).

In this study, we focus on the mature stage of cyclones, i.e when cyclones attain their maximum intensity. In earlier stages of their lifecycles, the forcing from the PV streamer will only be more important since baroclinic instability is the main

cyclogenesis processes. Nevertheless, the contribution of high PV values in the lower troposphere to cyclone development during the cyclogenesis stage cannot be neglected. Indeed, the cyclogenesis of several medicane cases—along with other cases—coincided with, or came after, major convective events (Miglietta et al. 2013; Flaounas et al. 2015; Fita and Flaounas, 2018). As discussed in section 4, the memory effect of PV tracers may be important for the intensification of cyclones even after the peak of convective activity.


While a cyclone develops, increasing values of diabatically produced PV in its centre may eventually lead to tropical transition, i.e. when a cyclone initially formed as a baroclinic low is eventually only sustained by latent heat release (Davis and Bosart, 2004). Our results show that this transition cannot be excluded in the Mediterranean; however, this seems to be a rather rare case scenario, especially due to the short lifespan of cyclones and the sharp land-sea transitions.  Such cases

would be outliers and found towards the top left of the scatterplot shown in Fig. 10. However, Fig. 10 shows that, for the cases considered here, these outliers were not identified as medicanes, probably due to not complying with the phenomenological criterion of a cloudless "eye" in their centre. Therefore, a new definition of medicanes that resides in atmospheric dynamics should be established as systems that are sustained, or strongly driven, by diabatic processes.

## 6. Summary and conclusions

From the PV-thinking perspective, a cyclone may be conceptualized as the amalgamation of different PV anomalies. In this framework, we implemented into WRF, and applied, an online diagnostic to 100 simulations of the most intense systems that occurred during a 10-year period, from 2008 to 2017. This diagnostic decomposed PV into different components, each related to a parametrized physical process, and enabled piecewise PV inversion to provide further insights into the role of different processes in cyclone development.


Our first objective aimed at quantifying the relative contribution of different processes to the PV-structure of Mediterranean cyclones. The vertical PV structure of the 100 cyclones showed two main contributions: one in the upper troposphere, related to the intrusion of the PV streamer into the Mediterranean, and one in the lower troposphere, related to the net production of PV due to diabatic processes. While the upper-tropospheric anomaly is almost solely composed of the conserved PV tracer

($q_{co}$), the one in the lower troposphere is the outcome of the sum of several contrasting, non-conserved PV tracers. More precisely, latent heat release is shown to correspond to a net source of PV from ground level until the mid troposphere,

approximately at 500 hPa. This net PV production was shown to be partly counteracted by sink processes related to longwave radiation and turbulent diffusion of temperature. Consequently, $q_{mp}+q_{cu}$ and $q_{lw}+q_{bl}$ present two vertical profiles almost symmetric about zero. Eventually, their sum corresponds to a positive PV anomaly at around 850 hPa, i.e. the level
where cyclones typically have their maximum intensity in terms of relative vorticity. Momentum mixing and frictional forces were shown to have an ambiguous effect on cyclone intensity. Indeed, $q_{mo}$ was shown to present both negative and positive signs in the lower troposphere though its average contribution over all 100 cyclones was shown to be positive. Interestingly, several cases presented exceptionally high $q_{mo}$ values, comparable to the PV produced due to latent heat release. Finally, our results showed that $q_{co}$ in the lower atmospheric levels may attain high values close to the mountains.
Such values pre-exist cyclogenesis and are expected to enhance the amplitude of the local maximum of PV in cyclone centres at 850 hPa, but also to balance negative PV values in the first model levels due to $q_{lw}+q_{bl}$.

Our second objective aimed at analysing cyclone development with regards to their baroclinic and diabatic forcing. In fact, Fig. 9 showed that there is hardly a straightforward relationship between the positive low-level PV anomalies due to latent
heat release ($q_{mp}+q_{cu}$) and cyclone intensity. Consequently, diabatic processes alone do not function as an index of cyclone intensity. Indeed, piecewise PV inversion showed that most cyclones developed through the contribution of both baroclinic and diabatic forcings. In fact, Fig. 10 showed that the two forcings tend to have a negative, linear relationship: the more a cyclone is diabatically driven, the less  baroclinic instability contributes to its mature stage. Piecewise PV inversion is a method that incorporates uncertainties due to technical challenges in its application. Therefore, our results distinguish the
processes that play primary roles in cyclone development, rather than quantify their precise contribution. In this context, several exemplary cases of Mediterranean cyclones were analysed using inverted conserved and non-conserved PV tracer fields. In most cases, the contribution of the upper PV streamer was shown to be important for the surface cyclone. In fact, cyclone centres were typically located in the centre, or within, the low-level cyclonic circulation imposed by $q_{co}$. A few cases were shown to be mainly driven by diabatic processes. In these cases, the PV streamer was still present, but imposed a rather
weak low-level cyclonic circulation. Other exemplary cases of Alpine cyclogenesis along with the case presented in Fig. 16b showed that the role of $q_{mo}$ is not negligible and may even play a primary role in sustaining the cyclonic circulation. Finally, we focused on medicanes and, in particular, on the processes that drive their development. Results showed that most of the known medicane cases developed due to the forcing of both diabatic and baroclinic processes, although some cases like Trixie were shown to have developed under the influence of a strong diabatic forcing.


This study sheds some light on the variable contribution of different atmospheric processes to the development of Mediterranean cyclones. In this regard, PV, and more precisely PV tracers, is a perfectly adequate method to translate processes into cyclone dynamics. However, PV is a complex variable and several methodological challenges are met when it is used for such analyses. Consequently, future work could use improved implementations of piecewise PV inversion that are
more consistent with the primitive equations of the model and more suitable for fine scale grids (e.g. Decker, 2010). In

addition, other case studies that are mainly driven by $q_{mo}$ should be investigated. It would come as a surprise if such cases are only found in the Mediterranean region. Therefore, an analysis using PV tracers should be also applied to a wide variety of other cyclones within the storm tracks. Such an effort should include the application of piecewise PV inversion to a much wider range of cyclones than the 100 cases used in this study. This will enhance our insights into the process-based continuum of cyclones, as shown in Fig. 10, and would likely reveal a wide number of diabaticaly driven cyclones. While focusing on the most intense cyclones is a reasonable approach to understand the main intensification processes of Mediterranean cyclones less strong cyclones are also relevant to high impact weather, especially concerning the occurrence of heavy rainfall (Flaounas et al. 2018). In subsequent studies, we thus aim to extend our analysis to more cyclones to understand their driving processes but this time from the perspective of impacts. Finally, it is in the perspectives of this study to provide a new definition of medicanes, based solely on atmospheric dynamics: Mediterranean cyclones that are sustained, or strongly driven, by diabatic processes. This definition would allow to the community to expand its focus to a broader group of cyclones than the limited number of cases, currently qualified as medicanes.

Code availability: The PV-tracers diagnostic corresponds to a single module, written in Fortran and its implementation requires additional modifications to several parts of the original WRF code. The whole PV-tracers package is adapted to WRF version 4.0 and might demand minor, additional modifications for its implementation in subsequent versions of the model. The diagnostic and all relevant source code modifications are freely available upon request.

**Acknowledgements** We are grateful to Baruch Ziv and to two anonymour Reviewers for their constructive comments and suggestions that greatly improved this article. EF is thankful to Oscar Martinez-Alvarado for his help and support while developing the PV budget diagnostic for WRF and to John Methven for interesting discussions on the interpretation of PV tracers.

**Financial support** EF acknowledges funding through the H2020 European Research Council (INTEXseas (grant no. 787652)). FT acknowledges funding through the German Research Foundation Grant RI 1771/4-1 and the Transregional Collaborative Research Center SFB / TRR 165 "Waves to Weather" (www.wavestoweather.de) funded by the German Research Foundation (DFG).

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

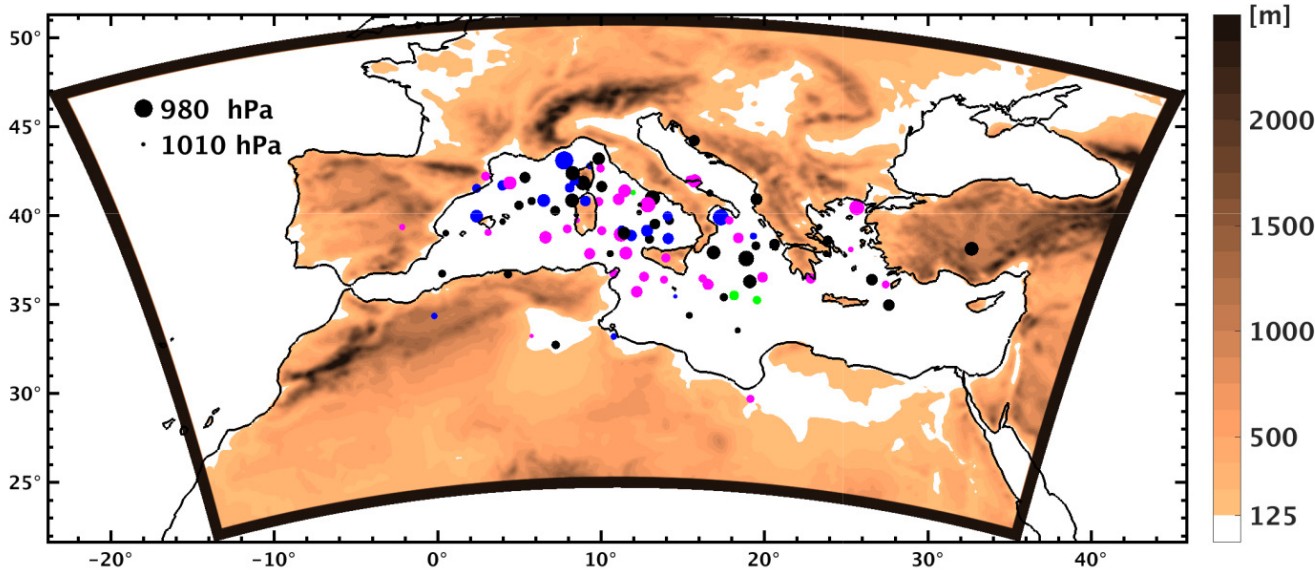

Figure 1: Simulations domain and orography (*colour*). *Dots* depict the locations of the 100 cyclones at their mature stage. The size of the dots is proportional to the cyclones' central mean sea level pressure (see legend upper left corner of the domain). Dot colours depict a certain season: *black* for winter, *magenta* for spring, *green* for summer and *blue* for autumn.

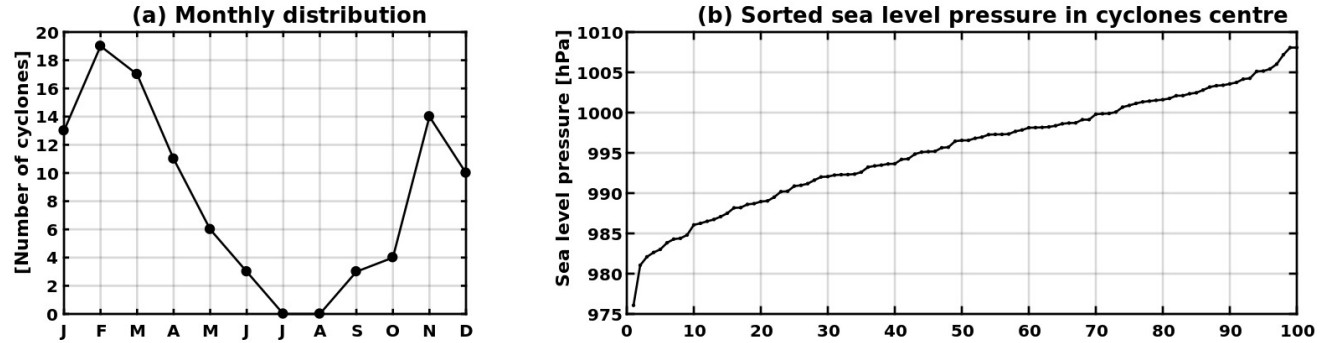

Figure 2: (a) Monthly distribution of cyclone occurrence. (b) Cyclones sorted according to their central mean sea level pressure during their mature stage.

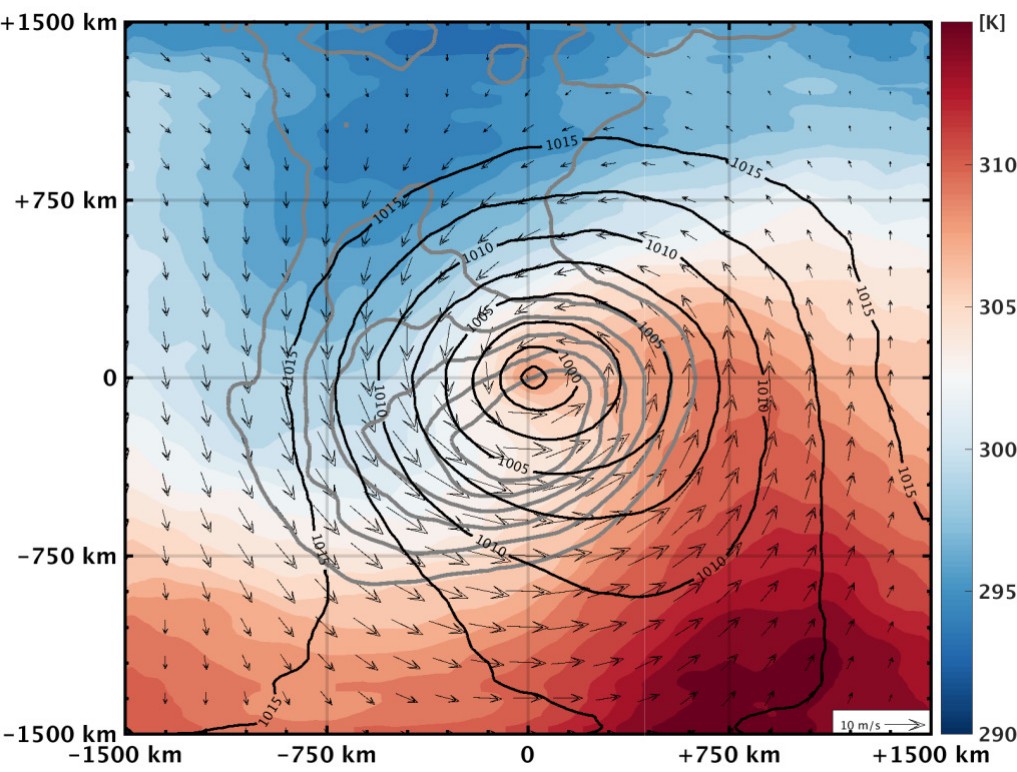


**Figure 3: Composite for all 100 cyclones at the time of their mature stage of PV at 300 hPa (in *grey contours* with 0.5 PVU contour intervals, starting from 1.5 PVUs), mean sea level pressure (in *black contours*) and horizontal wind (*arrows*) and equivalent potential temperature at 850 hPa (*colours*) with (0,0) denoting the position of cyclone centres.**


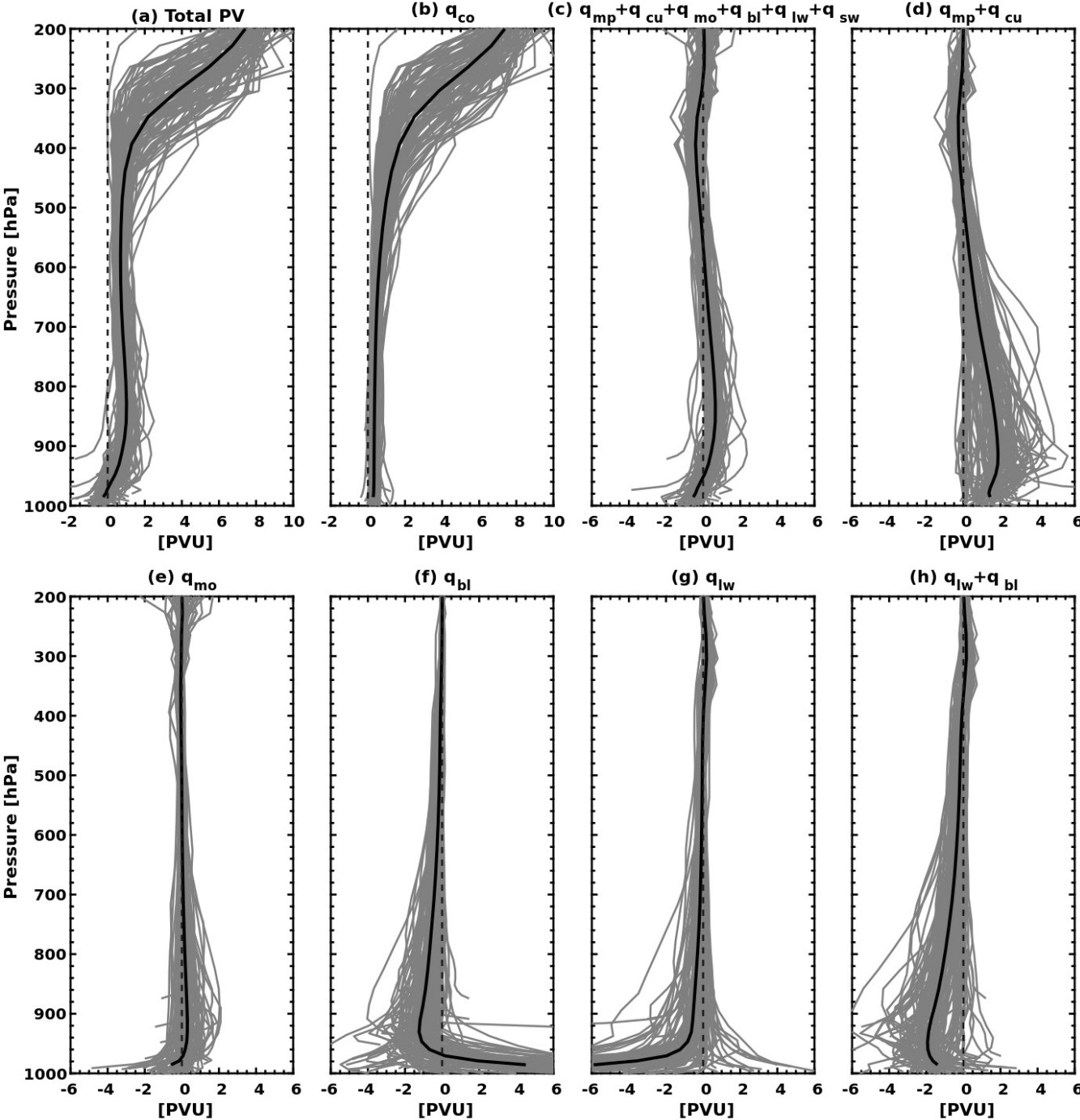

**Figure 4: Vertical profiles of total PV and PV tracers for all 100 cyclones (in *grey lines*), averaged 100 km around cyclone centres at the time of their mature stage. Profile averages and zero PVU are shown in *solid* and *dashed* black lines, respectively. Note that range of values in x-axis is not common for all panels.**

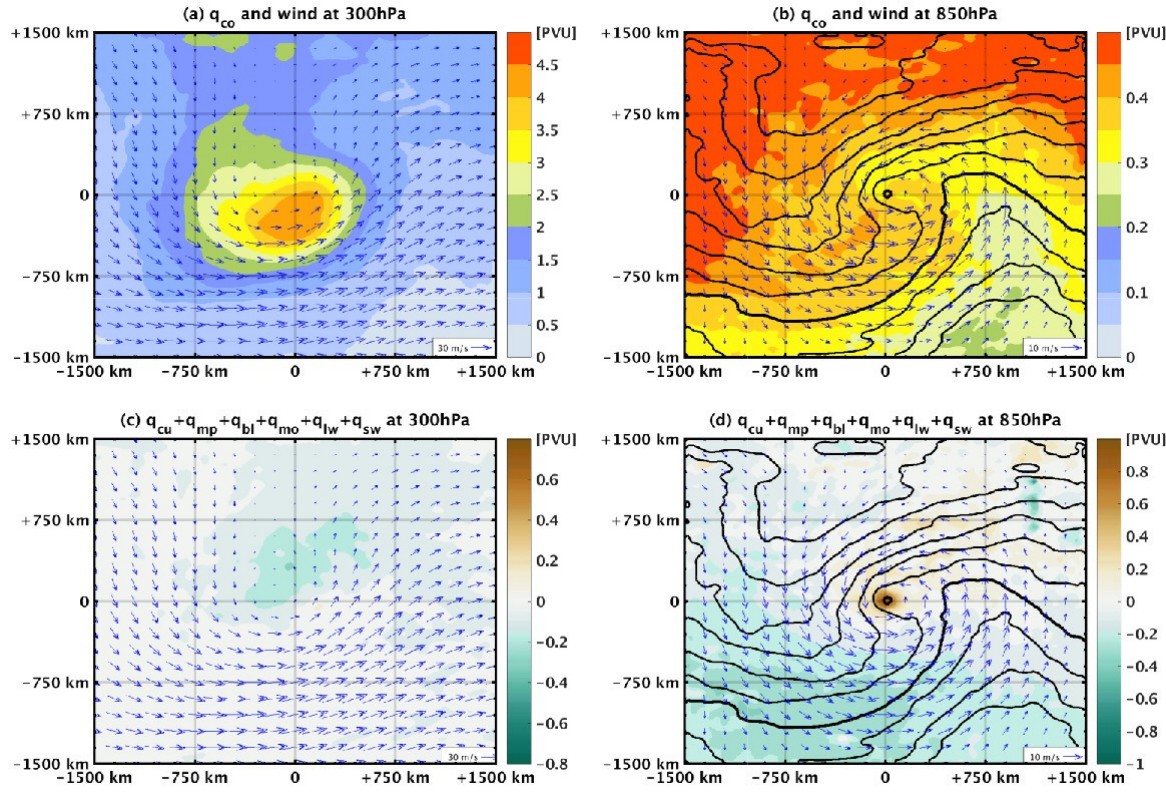


**Figure 5: (a) Composite averages for all 100 cyclones at the time of their mature stage of** $q_{co}$ **(in** *colour***) and wind (in** *arrows***) at 300 hPa. (b) as in a, but at 850 hPa. (c) as in (a), but for the sum of all non-conserved PV tracers. (d) as in (b) but for the sum of all non-conserved PV tracers. Note that the value ranges in the colourbars differ among panels. As a guide to the location of fronts, the composite average of equivalent potential temperature (**$\theta_e$**) at 850 hPa is also shown in panels (b) and (d) (***black contours***, every**
**2 K,** *bold* **for 308 K).**

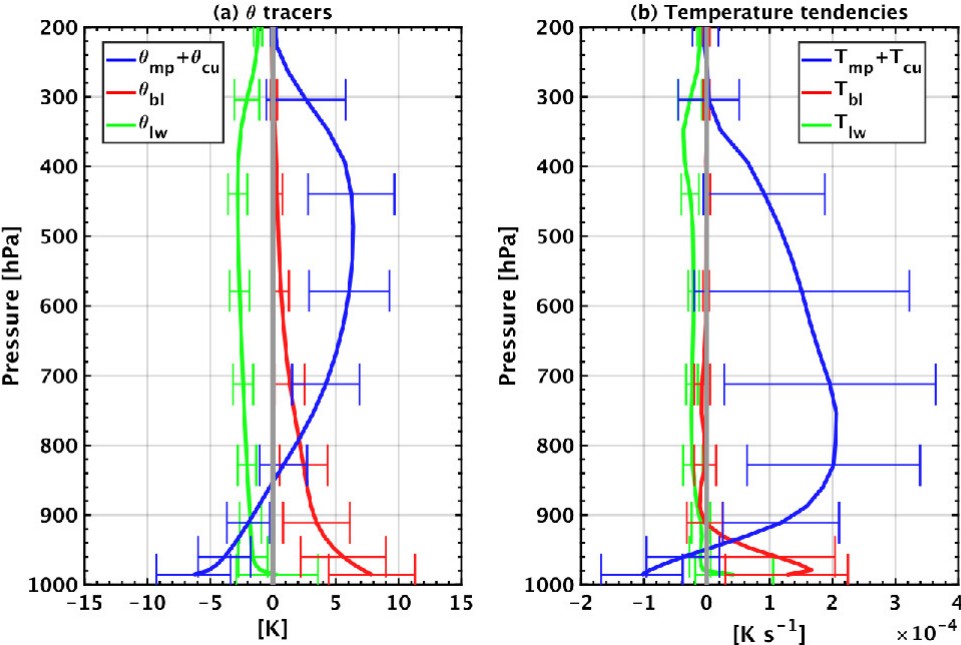

**Figure 6: (a) Average vertical profiles of $\theta$ tracers for all 100 simulations, averaged within a 100 km radius around cyclone centres; whiskers length equals twice the standard deviation. (b) as in (a) but for instantaneous temperature tendencies.**


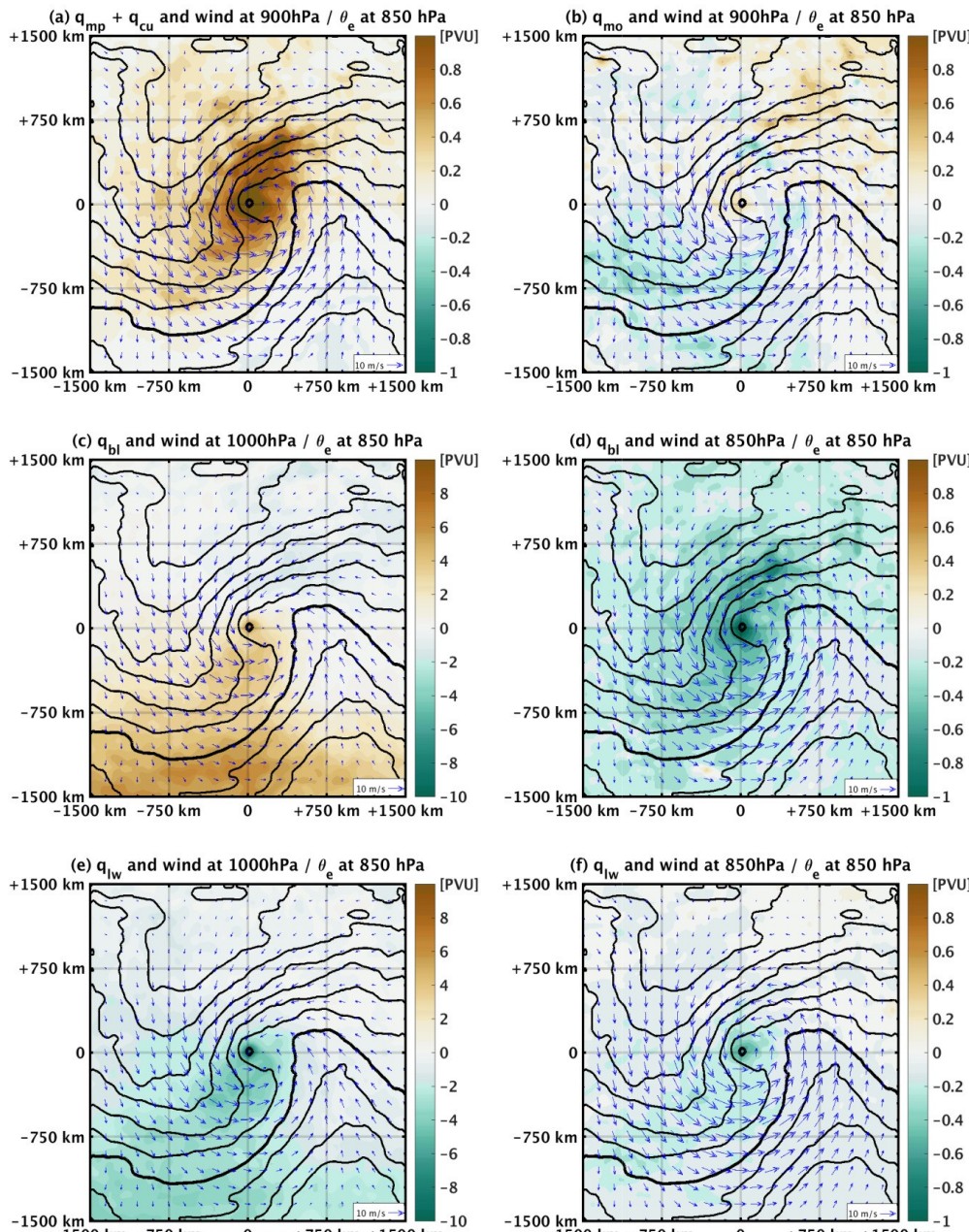

**Figure 7: Composite for all 100 cyclones at the time of their mature stage of horizontal wind (*arrows*) and of different PV tracers at different atmospheric levels (*colours*), with (0,0) denoting the position of the cyclone centres. As a guide to the location of fronts, the composite average of equivalent potential temperature ($\theta_e$) at 850 hPa is also shown (*black contours*, every 2 K, *bold* for 308 K). Note that the range of the colourbar values changes among panels.**

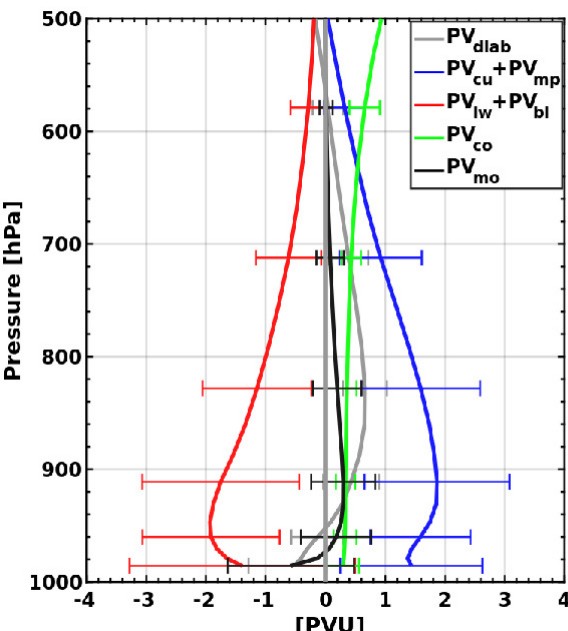

**Figure 8: Average vertical profiles of PV tracers for all 100 simulations, averaged within a 100 km radius around the cyclone centres. Whiskers length equals twice the standard deviation.**

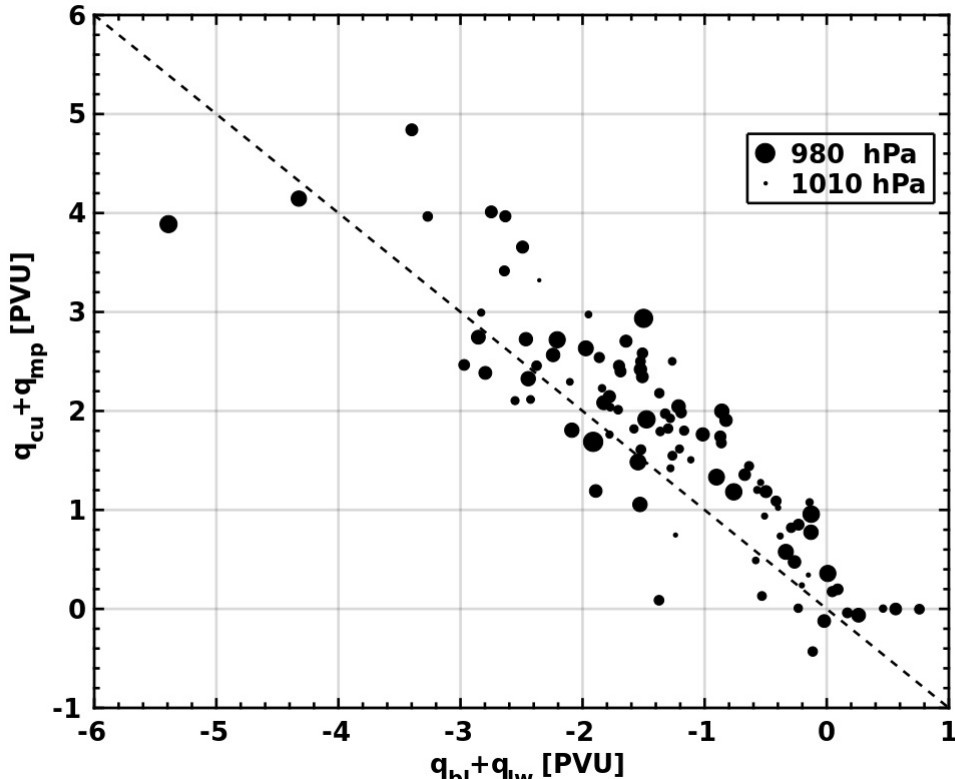

**Figure 9:** Scatterplot relating the sum of $q_{bl}$ and $q_{lw}$ to the sum of $q_{cu}$ and $q_{mp}$ for all 100 cyclones. Values of the PV tracers correspond to averages within a 100 km radius around the cyclone centres at the 850 hPa level. The size of the dot is proportional to the mean sea level pressure at the cyclone's centre and *dashed line* represents the $q_{cu}+q_{mp} = -(q_{bl}+q_{lw})$ equation.

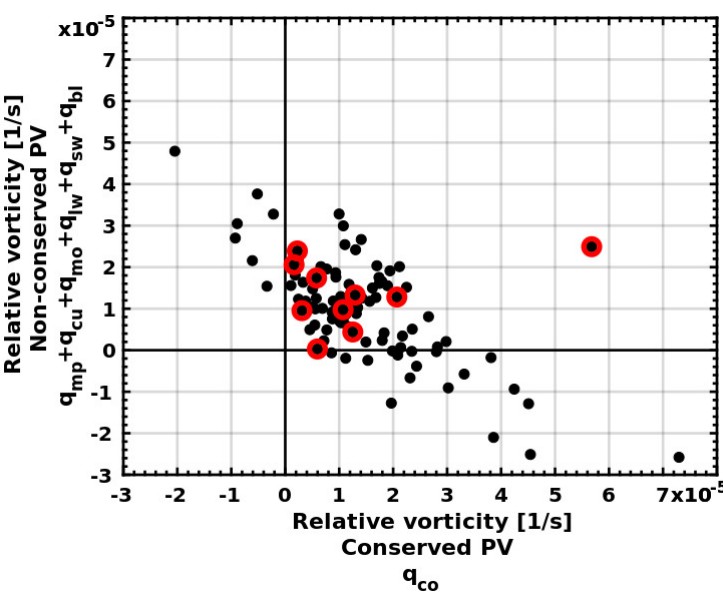

**Figure 10: Scatterplot relating the balanced relative vorticity fields after inverting the conserved PV tracers (x-axis) to the sum of all non-conserved PV tracers (y-axis).** *Red circles* **depict the ten medicane cases in our dataset.**




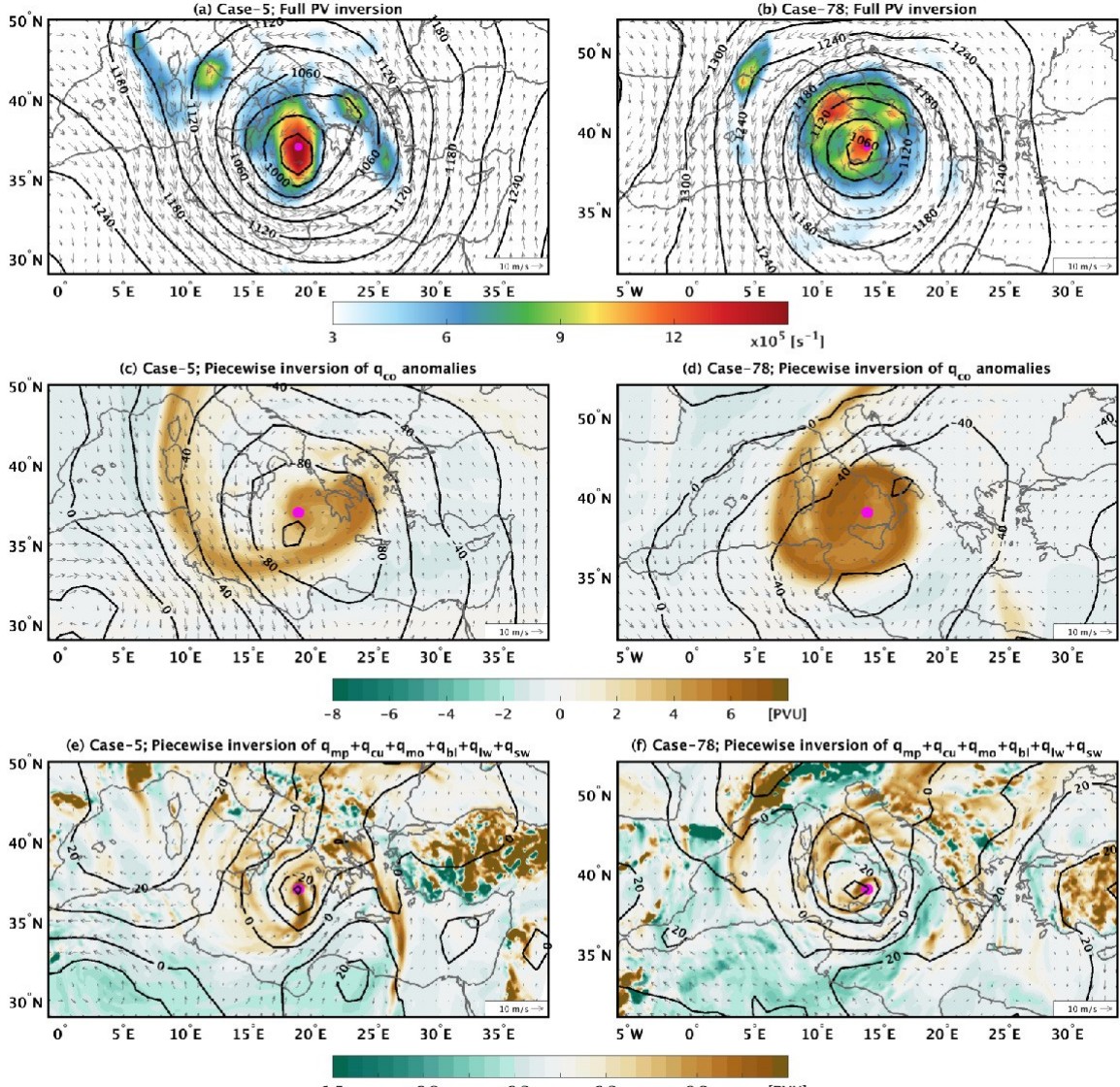

**Figure 11: (a)** Balanced relative vorticity (*in colour*), wind field (*in arrows*) and geopotential height (*in contour*) at 850 hPa after inverting the total atmospheric PV for cyclone case #5. The model outputs for this case are shown in the supplementary material. **(b)** as in (a), but for the case #78. **(c)** the field of $q_{co}$ at 300 hPa (*in colour*) and the balanced wind (*in arrows*) and geopotential height (*in contour*) fields at 850 hPa after inverting $q_{co}$. **(d)** as in (c) but for case #78. **(e)** the field of the sum of all non-conserved PV tracers at 850 hPa (*in colour*), as well as the balanced wind field (*in arrows*) and geopotential height (*in contour*) at 850 hPa after inverting all non-conserved PV. **(f)** as in (e), but for case #78. In all panels, the *magenta dot* depicts the cyclones' centre.

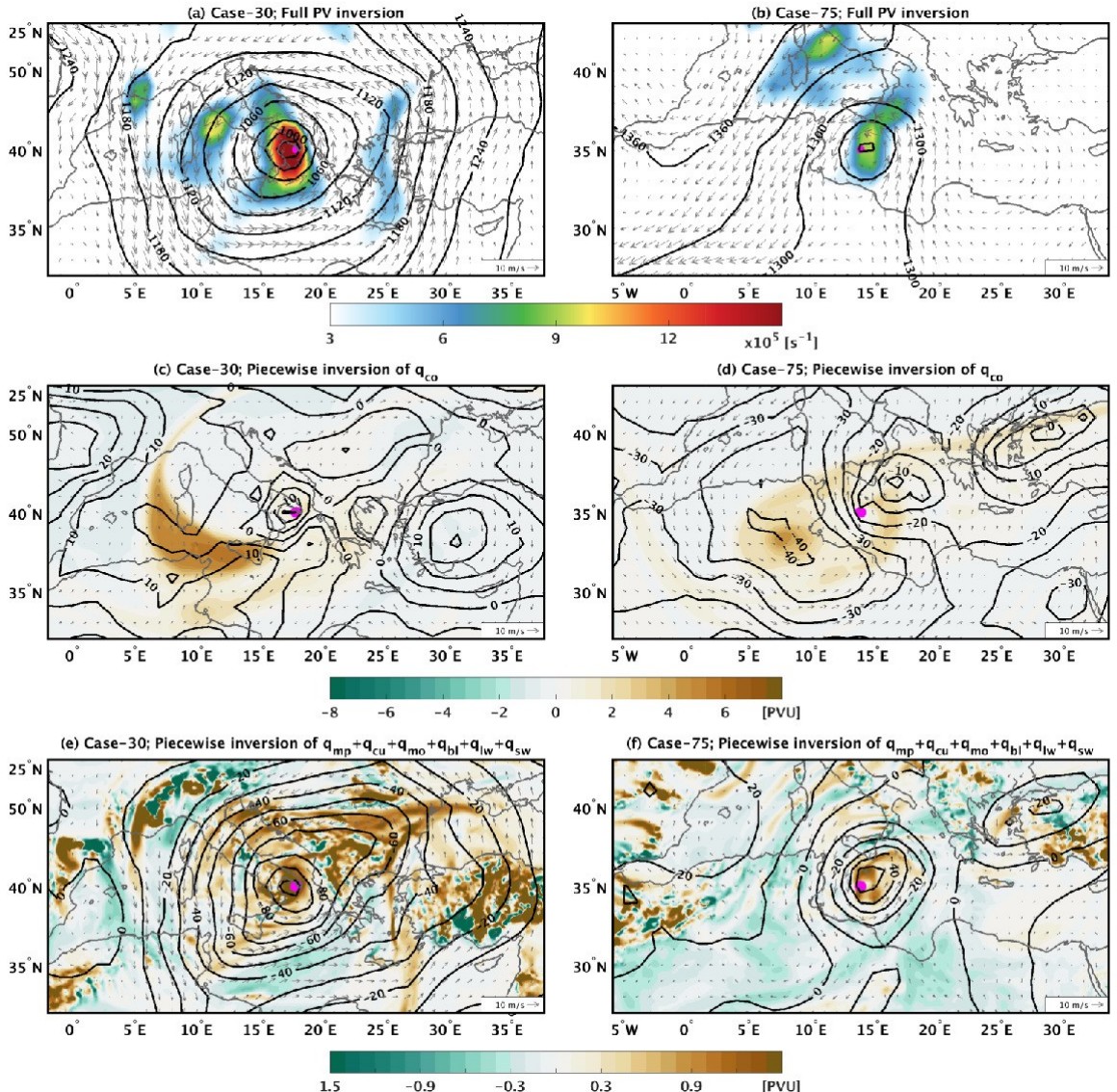

**Figure 12: As for Fig. 11 but for cases #30 and #75.**


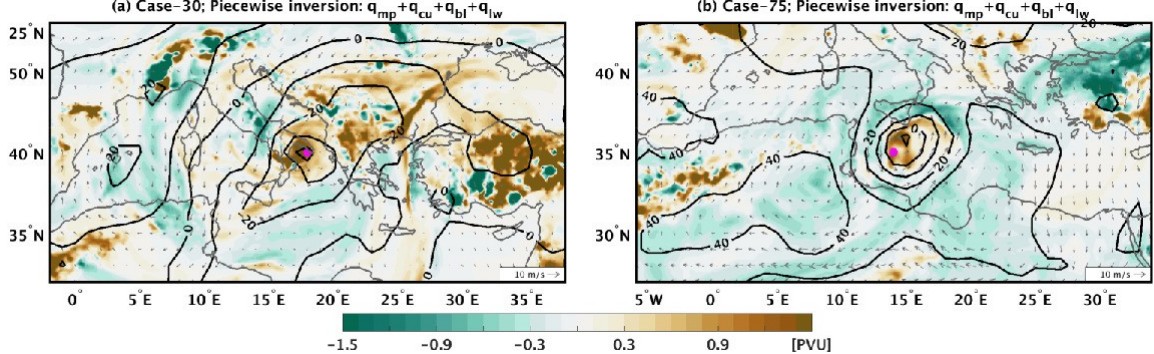

**Figure 13: (a)** The field of $q_{mp}+q_{cu}+q_{bl}+q_{lw}$ at 850 hPa (*in colour*) and the balanced wind (*in arrows*) and geopotential height (*in contours*) fields at 850 hPa after inverting $q_{mp}+q_{cu}+q_{bl}+q_{lw}$ for the cyclone case #30. **(b)** as in (a), but for case #75.

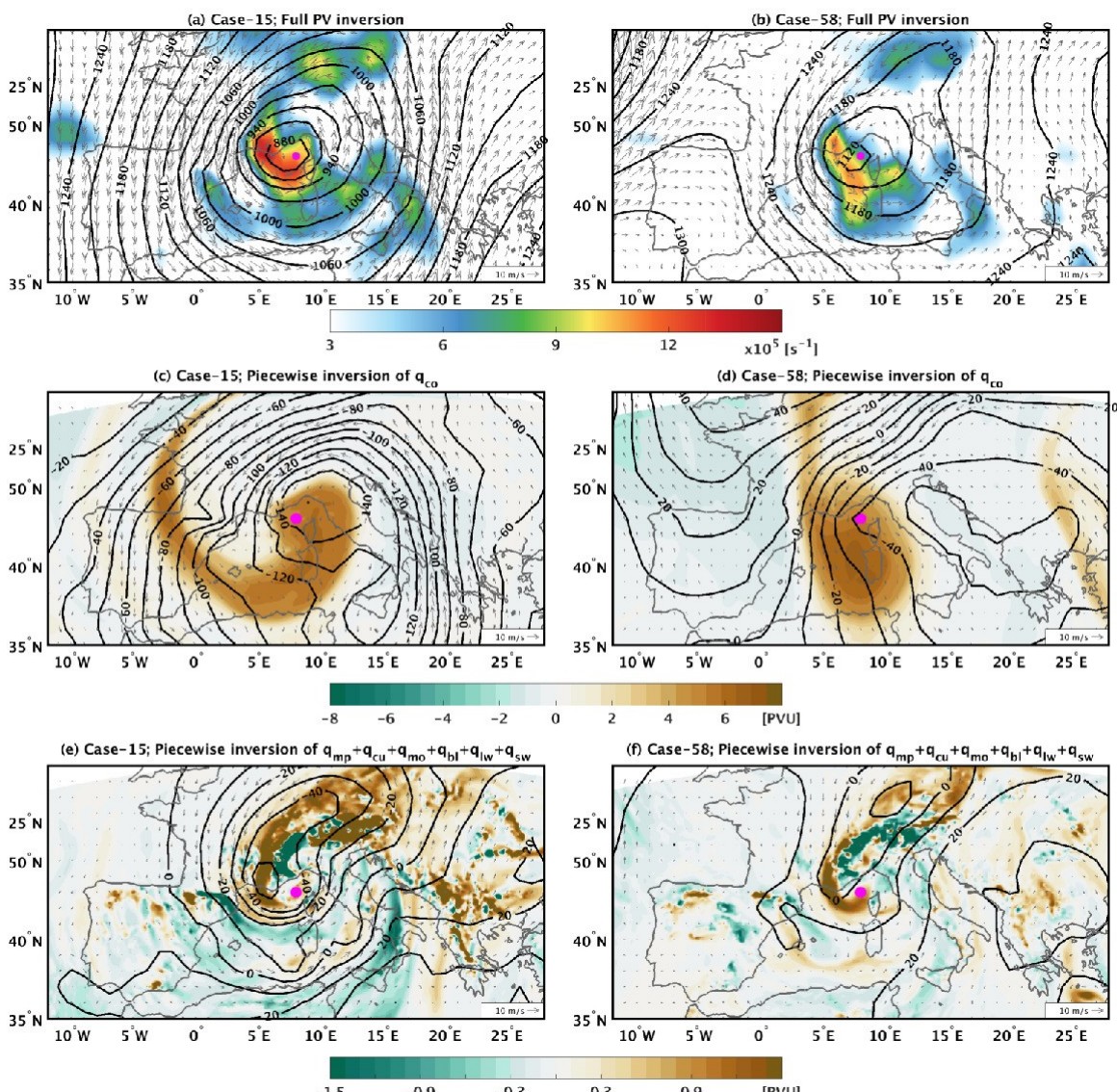

**Figure 14: As for Fig. 11, but for cases #15 and #58.**


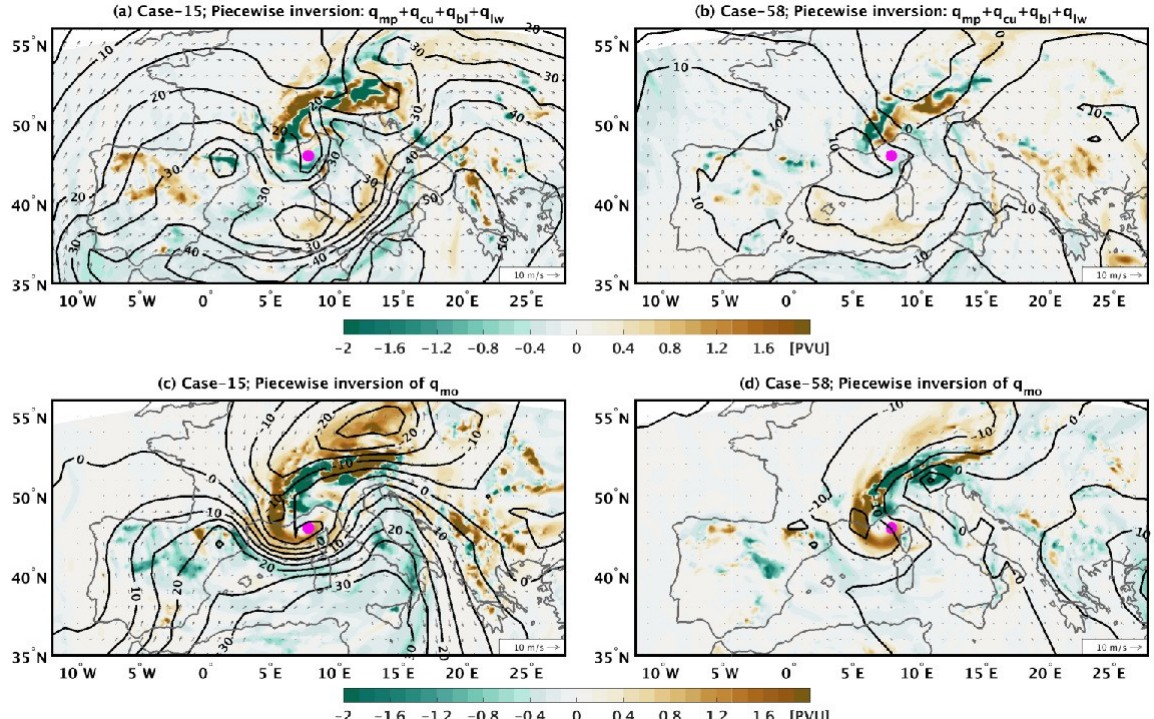

**Figure 15: (a) The field of** $q_{mp}+q_{cu}+q_{bl}+q_{lw}$ **at 850 hPa (*in colour*) and the balanced wind (*in arrows*) and geopotential height (*in contours*) fields at 850 hPa after inverting** $q_{mp}+q_{cu}+q_{bl}+q_{lw}$ **for the case #15. (b) as in (a), but for case #58. (c) the field of** $q_{mo}$ **at 850 hPa (*in colour*) and the balanced wind (*in arrows*) and geopotential height (*in contours*) fields at 850 hPa after inverting** $q_{mo}$, **for the case #15. (d) as in (c), but for case #58.**

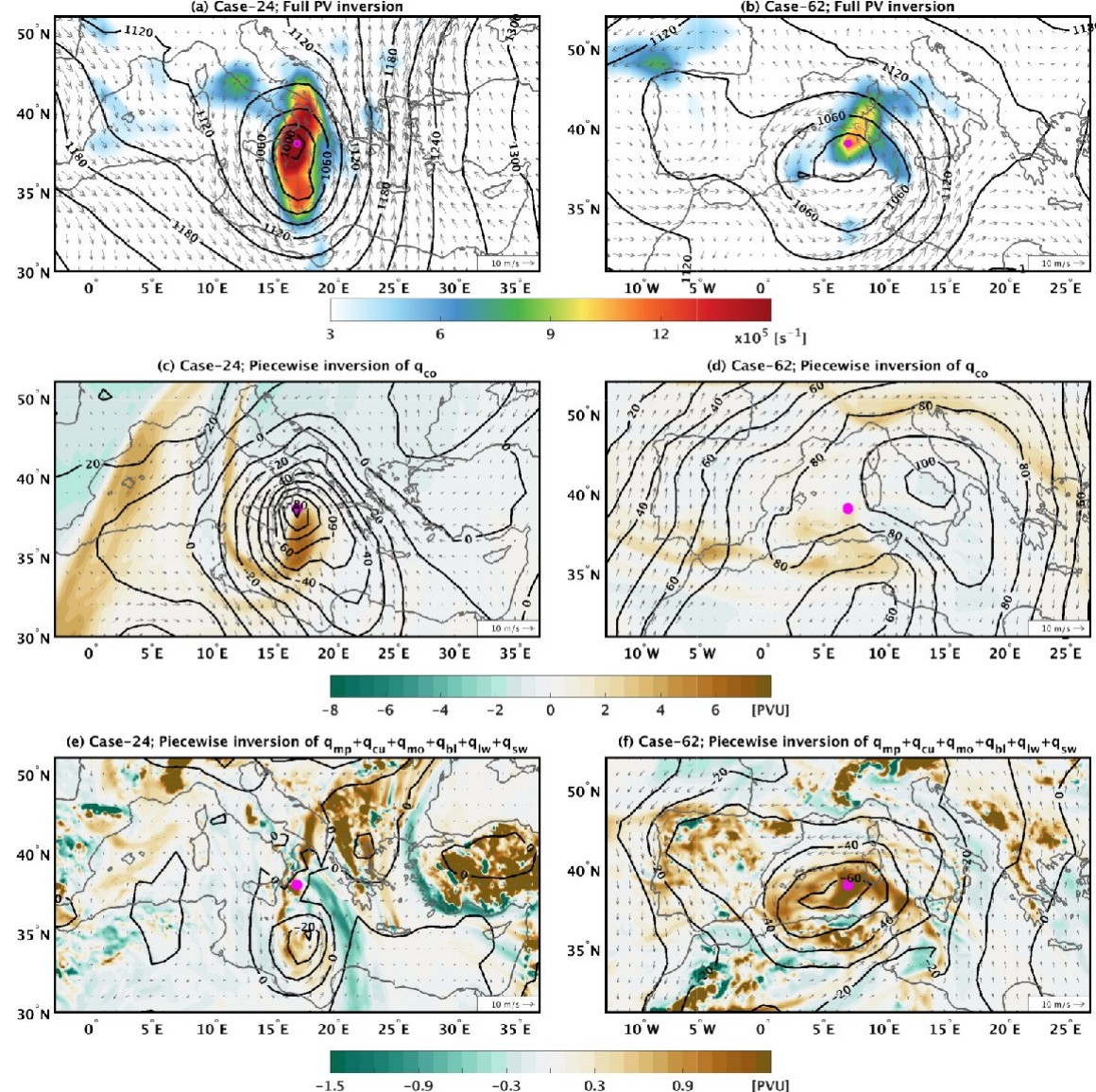

**Figure 16: As for Fig. 11, but for cases #24 and #62.**

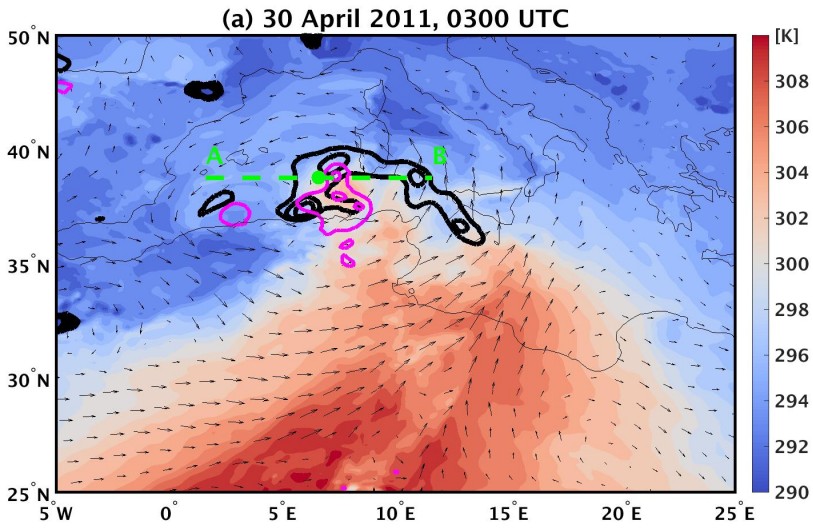

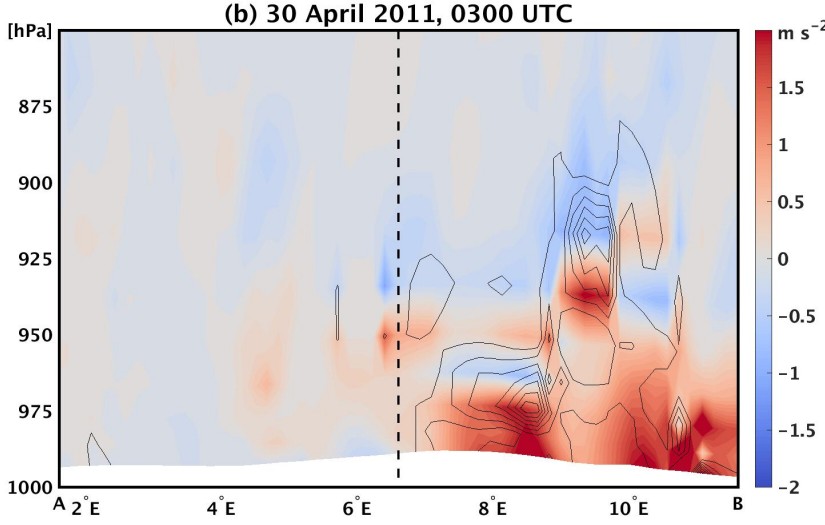

Figure 17: (a) fields of θ (in colour), wind (in arrows), $q_{mo}$ (in black contour with 0.5 PVU contour intervals, starting from 1 PVU) and $q_{mo1}$ (in magenta contour for -0.5 and -1 PVUs) at 850 hPa at the time of the mature stage of cyclone #62. (b) Vertical cross section along the green line of panel (a), showing in colours the field of $Fx$ (momentum forcing from PBL parametrisation) and in black contours the field of baroclinic PV production (sum of two last terms in Eq. 8), starting from 0.05 PVU s$^{-1}$ with an interval of 0.1 PVU s$^{-1}$.