# Peer review of "A process-based anatomy of Mediterranean cyclones: From baroclinic lows to tropical-like systems"

_Weather and Climate Dynamics, 2020_

## Referee Comment (RC2)

7 December 2020

Dear Editor,

Subject: Review of MS No.: wcd-2020-48

Following is my review of the manuscript "A process-based anatomy of Mediterranean cyclones: From baroclinic lows to tropical-like systems" by E. Flaounas, S. L. Gray amd F. Teubler.

The manuscript presents a comprehensive quantitative study of the dynamic and thermodynamic factors involved in the evolution 100 most intense Mediterranean cyclones during 2008-2017. The authors analyze the PV tracers representing the various cyclogenetic factors and evaluate their contributions through PV inversion. They show the interplay of the upper troposphere PV anomaly, related to the baroclinic forcing of cyclone development, as the major factor and the lower tropospheric diabatic latent heating as the secondary. They demonstrate various evolution scenarios through 10 selected cases. They devote a special section to Medicanes and show that, unlike their tropical counterpart, they develop under the influence of both baroclinic and diabatic processes.

I find the study **highly important**, and **strongly recommend publishing it**, after considering several comments specified below. There is no need for further review.

| line | Item | Comment |
| --- | --- | --- |
| 93-94 | "..and then sections 3 and 4 present our main results. Finally, section 5 hosts .. | Change the number "3" to "4" and so on. |
| 161 | Setting the initial conditions 36 hours before the cyclone's mature stage… | It would be of help if you specify the average rate and its STD of the vorticity increase during that time. |
| 189 | " .. $q_{co}$ anomalies have been defined as differences from three-day averages of $q_{co}$, i.e. the duration of each simulation." | In the Mediterranean region, there are cyclones that are slow moving, In such a case the 3 days average may be anomalous with respect to climatology and your signal may be found too weak. Refer to this point. |
| 272 | mountains plausibly act as a reservoir of relatively high PV | Change "reservoir" to "spurce" |
| Fig. 5 | b, d | What do the contours mean? |

I take this opportunity to urge the authors to extend their research beyond this paper, and to inquire the following issues:

1. The emphasis given to the most intense cyclone is reasonable from the energetic point of view. But, since the weather in the Mediterranean, and the rain in particular, may be severe also when the cyclones are not extremely intense, it would be of benefit to extend the analysis to less intense cyclones.
2. The diabatic heating, which was found second most important cyclogenetic factor here, is expected to depend on the SST, which has a seasonal distinct course. I suggest you to inquire the dependence of this factor on the time of the year.
3. The baroclinic factor is influenced by the topography surrounding the Mediterranean. Hence you may find a geographical variability in the contribution of this factor in the various parts of the Mediterranean.

Baruch Ziv

---

## Referee Comment (RC1) · Anonymous Referee #1 · 6 Oct 2020

Title: A process-based anatomy of Mediterranean cyclones: From baroclinic lows to tropical-like systems Authors: Flaounas et al. RECOMMENDATION: Minor revisions

The mature stage of 100 intense Mediterranean cyclones is analyzed by means of PV budget diagnostic and piecewise PV inversion applied to WRF model simulations. In this way, the relative contribution of different processes leading to cyclone intensification is investigated. Results show that the cyclones mostly develop due to a combination of diabatic and baroclinic processes. The paper is very well written and organized; e.g., the authors were able to explain the Equations in Section 2.2 in an effective way. Results are relevant for the topic of Mediterranean cyclones, and in particular they

shed new light on the mechanisms of development of Medicanes from a PV perspective. Thus, I recommend its publication almost in the present form. A few minor points are indicated below.

Line 100: "subjectively excluding cyclonic circulation features that did not present clear enclosing mean sea level pressure contours": could you identify a more objective method? Then, why Case-75 (See Supplementary material) has been included?

Line 110: 34 vertical levels are below standard: did you check if the number of levels affected the simulation results?

Line 120: please, provide a reference to Eqs. (1) and (2);

Line 121: the Coriolis parameter is not shown explicitly in the Equations;

Line 191-192: if my interpretation is correct, I suggest to rephrase as: "has already been applied in past studies to upper-tropospheric PV anomalies in order to separate the advective contributions of upper-tropospheric and low-tropospheric PV and potential temperature anomalies";

Line 219: I do not see many cases in the eastern Mediterranean;

Line 231: do you mean 1000-1005 hPa, 995-1000 hPa and 985-990 hPa?

Figures 1-3: are the fields extracted from ERA-5 or are WRF model outputs?

Line 326: 5d -> 5c

Line 337: you may consider (it is not necessary) to show the expansion of Eq. (4) in three terms in an Appendix;

Line 484: 16c -> 16b

Line 590-591: while the effort of the Authors to provide a definition of Medicanes is appreciable, no quantitative criteria are provided, which indicate that additional studies should be performed to reach this task.

---

## Referee Comment (RC3) · Anonymous Referee #3 · 27 Dec 2020

This study presents an in-depth and thorough analysis of Mediterranean cyclones from a potential vorticity (PV) perspective. The authors use a very rigorous approach by computing an on-line PV budget in the Weather, Research, and Forecasting (WRF) model, along the lines of Stoelinga (1996). The study presents a large number and variety of cases derived from an ERA5 climatology, and simulated with WRF to collect PV information associated with various physical processes (including physics parameterizations). Then, PV inversion was used to quantify the contributions of different physical processes to the cyclone's circulation. Results demonstrate the varying contributions of different processes, including "Medicane" systems that are shown to rely heavily, but not always exclusively on diabatic processes. The conclusions and interpretations in

this study are strongly supported by rigorous analysis and clear graphics. In sum, I review many papers, and this is one of the stronger papers I've reviewed recently. I enjoyed reading it.

However, there are ways that the study could be strengthened and clarified, though I view these suggestions as minor revisions.

In terms of the WCD review criteria, I would rate the Scientific significance as "good" (see main suggestion #1 below), Scientific quality as "excellent", and Presentation quality is good to excellent.

Main suggestions: 1) While the goal and focus of the study is on Mediterranean cyclones, the results seem disconnected from other studies that have used PV-based methods to quantify cyclone dynamics in different geographical regions. I was surprised that even some of the PV-based cyclone classification studies that include the second author were not cited, for example (e.g. Gray and Dacre 2006). Also, none of Ahmadi-Givi et al (2004), Deveson et al. (2002), Plant et al. (2003) were cited. Placing your Mediterranean results within the context of existing literature using similar methods in different would strength your conclusion sections, where you allude to such connections. This would elevate the scientific significance of your study.

2) The process you call "momentum forcing" is unclear, and possibly incomplete. What, exactly, is this process? Can you please explain this more clearly, perhaps with a schematic? Also, you selected a convective parameterization scheme that doesn't adjust momentum (Kain-Fritsch), and so convective momentum adjustment is missing in your model runs. Perhaps the slack is picked up by the turbulence parameterization? It would be interesting to see if there would be differences in the online PV budget if you used one of the WRF schemes that also adjusts momentum (e.g., Zhang-McFarlane, Arakawa-Schubert, Tiedtke...)

3) There are some other classic Alpine lee cyclone studies of Mediterranean cyclones that are not cited, for example, especially Bleck and Mattocks (1984), who used a

PV-based approach to study this phenomenon:

Bleck, R., & Mattocks, C. (1984). A preliminary analysis of the role of potential vorticity in Alpine lee cyclogenesis. Beiträge zur Physik der Atmosphäre, 57(3), 357-368.

Minor corrections and suggestions: Line 40: "Topographic PV streamers" comes to my mind, but they are "banners" here. Are the terms used here consistent with previous literature? E.g., Aebischer and Schär (1998).

Line 49: Suggest adding modifier "Mediterranean" before "cyclones"

Line 67: It may be a minor, semantic point, but I would disagree that convection affects all cyclones, unless you are including slantwise convection perhaps. Upright convection is often, but not always present. Some cyclones, such as "heat lows" in the Southwestern US, can be accompanied by clear skies, albeit perhaps with dry convection during daytime hours. Some lee cyclones may be devoid of convection though.

Lines 110-112: I am curious how you implemented the on-line PV code with the WRF hybrid coordinate. The PV tendency equations would then have to vary with the vertical coordinate. This must have been very difficult, technically, and I am impressed. I would love to work with this module, as would many others I'm sure – is the code available to share?

Lines 113-115: Again, somewhat minor, but I would not recommend simulating tropical-like systems with a 5-class microphysics scheme. Without hail or graupel, snow is unrealistically lofted in strong, convectively forced systems with updraft speeds greater than a m/s or so.

Section 2.2, around line 170: Citation of some prior work may be useful here, such as the Schubert et al. 2004 paper: Schubert, W.H., Hausman, S.A., Garcia, M., Ooyama, K.V. and Kuo, H.C., 2001. Potential vorticity in a moist atmosphere. Journal of the atmospheric sciences, 58(21), pp.3148-3157.

Line 183: Check spelling on Stoelinga

Line 189: The 3-day time average is quite a lot shorter than Davis' 7-day average. No need to re-do anything, but what are the implications? My thought is that some upper-level PV contributions are lost, as the anomaly computed as a deviation from a 3-day average will be weaker for large, strong, persistent upper troughs.

Figure 1: Note typo in caption "smean ea"

Line 240: The position of cold-frontal, lower-tropospheric cyclonic PV maxima will vary from case to case, and so in a composite, there will be a large degree of cancellation (e.g., the type of feature highlighted by Lackmann 2002): Lackmann, G.M., 2002. Cold-frontal potential vorticity maxima, the low-level jet, and moisture transport in extratropical cyclones. Monthly Weather Review, 130(1), pp.59-74. This is also mentioned and discussed near lines 325-327.

Line 265: At some point, it seems that diurnal changes in PBL depth, and the fact that PV $\sim$ 0 in the PBL should be mentioned or discussed. Or perhaps better near line 360.

Line 285: I wonder if there is any seasonality to the varying contributions?

Line 458: Capitalize Gulf

Line 484: Is it Fig. 16c that should be referenced?

Line 506: Can references be provided to support this sentence?

---

## Editor Comment (EC1) · Nili Harnik (Editor) · 30 Dec 2020

All three reviewers are highly supportive of the work and recommend publication subject to some changes or comments. While the comments are referred to as minor, I urge the authors to respond to the comments in detail as this will improve the paper and its influence.

In particular, there are some comments regarding the equations and the modeling framework and its physical and numerical details (mostly by Rev 1 and Rev 3 - in particular comment 2).

Also, I agree with reviewer 3 that the authors should place their results in the wider context of the use of PV inversion and cyclone dynamics in other regions.

Both reviewers 2 and 3 comment about the possible seasonality of the varying contributions (reviewer 2 also comments about the geographic dependence)- for the sake of the open discussion, I urge the authors to respond to these points, especially to the explicit suggestions by reviewer 2, and if fitting, to add a comment to the discussion in the paper.

To facilitate the reading of the paper, I suggest adding a table with the different PV inversion variable definitions

---

## Author Response (AR1)

**Reply to anonymous Reviewer #1**

*Title: A process-based anatomy of Mediterranean cyclones: From baroclinic lows totropical-like systems Authors: Flaounas et al. RECOMMENDATION: Minor revisions*

*The mature stage of 100 intense Mediterranean cyclones is analyzed by means of PV budget diagnostic and piecewise PV inversion applied to WRF model simulations. In this way, the relative contribution of different processes leading to cyclone intensification is investigated. Results show that the cyclones mostly develop due to a combination of diabatic and baroclinic processes. The paper is very well written and organized; e.g., the authors were able to explain the Equations in Section 2.2 in an effective way. Results are relevant for the topic of Mediterranean cyclones, and in particular they shed new light on the mechanisms of development of Medicanes from a PV perspective. Thus, I recommend its publication almost in the present form. A few minor points are indicated below.*

We would like to thank the Reviewer for the positive review, for carefully reading our manuscript and for the fruitful comments.

*Line 100: "subjectively excluding cyclonic circulation features that did not present clear enclosing mean sea level pressure contours": could you identify a more objective method? Then, why Case-75 (See Supplementary material) has been included?*

Thank you for this comment. We are now more specific on our criterion to reject a cyclone track as a "bogus" one. We added the following in section 2:

*"Out of all tracked cyclones, we retained the 100 most intense ones. However, the tracking method is sensitive to identifying long lasting relative vorticity local maxima and in several cases these maxima lacked a clear cyclonic structure, i.e. they did not correspond to structured wind vortices. These tracked features were characterised as "false positives" and were subjectively excluded from our dataset of 100 most intense cyclones. Our main criterion to retain a system is the existence of a clear relative vorticity maximum that is outlined by enclosing mean sea level pressure contours."*

Case-75 was associated with rather large values of sea level pressure and therefore contours were not visible in the supplementary material. Supplementary material has been redone to include contours of larger values.

*Line 110: 34 vertical levels are below standard: did you check if the number of levels affected the simulation results?*

Thank you for this insightful comment. It is rather difficult to safely conclude on the number of vertical levels that brings "best practice" to cyclone simulations. For instance, Ma et al. (2012) showed that it is not the number of levels that has an impact on the simulation quality of a tropical cyclone, but their selective density within different atmospheric layers. In another case study of a cyclone, Lean and Clark (2003) performed several sensitivity tests on the model horizontal resolution and number of vertical levels. They concluded that a 45-level model is sufficient to represent the overall frontal structure of a cyclone in fine horizontal resolutions of 2 and 4 km. A 90-level model is though required to reproduce small-scale circulations. These results are fairly consistent with our choice of using 35 model levels in a spatial resolution of 15 km.

We agree with the Reviewer that L35 is close to the minimum number of vertical levels that we typically meet in the scientific literature, although quite common for WRF and other model simulations, even in late papers. For instance, Martinez-Alvarado et al (2014) analysed diabatic heating

rates in warm conveyor belts using the MetUM and COSMO models in resolutions of 12 and 14 km and 38 and 40 vertical levels, respectively.

We did not perform any sensitivity tests and we consider that such an exercise would escape the scope of this paper, especially when considering the high number of simulations performed for this study and the even more needed to perform the sensitivity tests per case (even if this will be done for few cases).

It is important to state that our priority in this paper is to qualitatively determine the driving processes that intensify Mediterranean cyclones and this is unlikely to be -strongly- dependent to the number of vertical levels that we use. Despite the importance of the Reviewer's query, we chose not to include additional comments in the manuscript to avoid confusion.

Lean, H.W. and Clark, P.A. (2003), The effects of changing resolution on mesocale modelling of line convection and slantwise circulations in FASTEX IOP16. Q.J.R. Meteorol. Soc., 129: 2255-2278. https://doi.org/10.1256/qj.02.57

Ma, Z., Fei, J., Huang, X., & Cheng, X. (2012). Sensitivity of tropical cyclone intensity and structure to vertical resolution in WRF. Asia-Pacific Journal of Atmospheric Sciences, 48(1), 67-81.

Martínez-Alvarado, O., Joos, H., Chagnon, J., Boettcher, M., Gray, S.L., Plant, R.S., Methven, J. and Wernli, H. (2014), The dichotomous structure of the warm conveyor belt. Q.J.R. Meteorol. Soc., 140: 1809-1824. https://doi.org/10.1002/qj.2276

*Line 120: please, provide a reference to Eqs. (1) and (2);*

We have rephrased part of the text to make it clear that Eqs 1 and 2 come from Cao and Xu (2011):

*"Equations 1 and 2 follow the formulation of Cao and Xu (2011) that has the advantage of avoiding wind and temperature interpolations—or approximations—in atmospheric levels other than the ones in model outputs."*

*Line 121: the Coriolis parameter is not shown explicitly in the Equations;*

Reference to the Coriolis parameter is now deleted.

*Line 191-192: if my interpretation is correct, I suggest to rephrase as: "has already been applied in past studies to upper-tropospheric PV anomalies in order to separate the advective contributions of upper-tropospheric and low-tropospheric PV and potential temperature anomalies";*

We rephrased as suggested.

*Line 219: I do not see many cases in the eastern Mediterranean;*

We rephrased to:

*"The cases tend to be concentrated over maritime areas, especially over the western Mediterranean Sea."*

*Line 231: do you mean 1000-1005 hPa, 995-1000 hPa and 985-990 hPa?*

Indeed, thank you for this correction. We rephrased as suggested.

*Figures 1-3: are the fields extracted from ERA-5 or are WRF model outputs?*

All figures present simulation results. This is now clearly mentioned.

*Line 326: 5d -> 5c*

Done.

*Line 337: you may consider (it is not necessary) to show the expansion of Eq. (4) in three terms in an Appendix;*

Also in accordance with the comments of the third Reviewer, Eq. 4 is now expanded and relevant discussion is enriched (see reply to the third Reviewer).

*Line 484: 16c -> 16b*

Done.

*Line 590-591: while the effort of the Authors to provide a definition of Medicanes is appreciable, no quantitative criteria are provided, which indicate that additional studies should be performed to reach this task.*

Thank you for this comment. The scatterplot in Fig. 10 shows that a classification of cyclones based on baroclinic vs diabatic processes yields a continuum of cyclones, rather than distinct clusters. It is therefore difficult to use quantitative criteria to identify medicanes as a well-defined, separate group of cyclones (at least from the perspective of diabatic vs baroclinic forcing to their development). Nevertheless, a population of 100 cyclones might not be enough to classify cyclones according to their driving processes. After all, our peculiar cases in Fig. 16 show that there are outliers that could potentially make part of distinct cyclone categories.

We reformulated our perspective studies in the last paragraph of the conclusions as follows:

*"Therefore, an analysis using PV tracers should be also applied to a wide variety of other cyclones within the storm tracks. Such an effort should include the application of piecewise PV inversion to a much wider range of cyclones than the 100 cases used in this study. This will enhance our insights into the process-based continuum of cyclones, as shown in Fig. 10, and would likely reveal a wide number of diabatically driven cyclones. While focusing on the most intense cyclones is a reasonable approach to understand the main intensification processes of Mediterranean cyclones less strong cyclones are also relevant to high impact weather, especially concerning the occurrence of heavy rainfall (Flaounas et al. 2018). In subsequent studies, we thus aim to extend our analysis to more cyclones to understand their driving processes but this time from the perspective of impacts. Finally, it is in the perspectives of this study to provide a new definition of medicanes, based solely on atmospheric dynamics: Mediterranean cyclones that are sustained, or strongly driven, by diabatic processes. This definition would allow to the community to expand its focus to a broader group of cyclones than the limited number of cases, currently qualified as medicanes."*

**Reply to Baruch Ziv**

*Following is my review of the manuscript "A process-based anatomy of Mediterranean cyclones: From baroclinic lows to tropical-like systems" by E. Flaounas, S. L. Grayamd F. Teubler. The manuscript presents a comprehensive quantitative study of the dynamic and thermodynamic factors involved in the evolution 100 most intense Mediterranean cyclones during 2008-2017. The authors analyze the PV tracers representing the various cyclogenetic factors and evaluate their contributions through PV inversion. They show the interplay of the upper troposphere PV anomaly, related to the baroclinic forcing of cyclone development, as the major factor and the lower tropospheric diabatic latent heating as the secondary. They demonstrate various evolution scenarios through 10 selected cases. They devote a special section to Medicanes and show that, unlike their tropical counterpart, they develop under the influence of both baroclinic and diabatic processes. I find the study highly important, and strongly recommend publishing it, after considering several comments specified below. There is no need for further review.*

We are grateful to Baruch Ziv for the positive review and the constructive comments.

*Line 93-94: "..and then sections 3 and 4 present our main results. Finally, section 5 hosts..". Change the number "3" to "4" and so on.*

Done.

*Line 161: "Setting the initial conditions 36 hours before the cyclone's mature stage...". It would be of help if you specify the average rate and its STD of the vorticity increase during that time.*

The initial conditions of the model simulation have been set 1.5 days before maturity of the cyclones as defined by the ERA5 tracks. However, when tracking cyclones in the 100 simulations, maturity is reached 1.9 days on average after the initial conditions, while cyclone tracks present their first track point about 9 hours after simulations' initial conditions. Inconsistencies are reasonably expected due to differences in the representation of cyclones between the model and the re-analyses and due to the fact that model outputs are produced with a 3-hour interval, compared to hourly fields in ERA5.

Cyclones' first track points present a relative vorticity average of $3.9 \times 10^{-5}$ s$^{-1}$. This value has to be compared with the average of $19 \times 10^{-5}$ s$^{-1}$ at cyclones' mature stage. This roughly yields an average intensification rate of $0.5 \times 10^{-5}$ s$^{-1}$ per hour. However, it is delicate to use these numbers to draw conclusions about cyclones intensification rate. This is due to our cyclone tracking method being sensitive to the high resolution of WRF simulations. Indeed, the average of $3.9 \times 10^{-5}$ s$^{-1}$ in cyclones' first track point is close to the threshold of $3 \times 10^{-5}$ s$^{-1}$, used by our tracking method to qualify a local maximum of relative vorticity as a cyclone centre. Since horizontal resolution defines the magnitude of the relative vorticity field at 850 hPa, it is arguable if the tracked features close to the timing of the initial conditions can be safely considered as the initial stages of a developing cyclone or local vorticity perturbations that extend the tracks in a rather unrealistic way. Studying cyclogenesis stage in high resolution simulations would demand prior adaptations to the cyclone tracking method. Such adaptations are not however necessary when considering cyclones' mature stage since the latter is representative of simulations' overall maximum of relative vorticity. Considering this discussion to be of very technical nature, we choose not to include it to the methods section.

*Line 189 In the Mediterranean region, there are cyclones that are slow moving, In such a case the 3 days average may be anomalous with respect to climatology and your signal may be found too weak. Refer to this point.*

Also in accordance with the comment of Reviewer #3, we added the following:

*"A three-days time period might be short to define PV anomalies in cases of slowly evolving upper tropospheric systems. However, anomalies need to be consistently defined for all 100 cyclones and Mediterranean cyclones are systems of relatively short lifespan. Therefore, we consider that a three-days average is a fair compromise for the application of piecewise PV inversion in this study."*

*Line 272 Change "reservoir" to "spurce"*

Changed reservoir to source.

*Fig. 5b, d What do the contours mean?*

We missed to include this information. As for Fig. 7, we added the following:

*"As a guide to the location of fronts, the composite average of equivalent potential temperature ($\theta_e$) at 850 hPa is also shown in panels (b) and (d) (black contours, every 2 K, bold for 308 K)."*

*I take this opportunity to urge the authors to extend their research beyond this paper, and to inquire the following issues:*

1. The emphasis given to the most intense cyclone is reasonable from the energetic point of view. But, since the weather in the Mediterranean, and the rain in particular, may be severe also when the cyclones are not extremely intense, it would be of benefit to extend the analysis to less intense cyclones.

Thank you for this comment. We fully agree with the Reviewer that it would be interesting to also understand the dynamics of less intense cyclones, but still capable of producing high impact weather. Extending our analysis to more cyclones in this article would be substantial amount of work. Given however the interesting suggestion, we included the following into the conclusions:

*"While focusing on the most intense cyclones is a reasonable approach to understand the main intensification processes of Mediterranean cyclones less strong cyclones are also relevant to high impact weather, especially concerning the occurrence of heavy rainfall (Flaounas et al. 2018). In subsequent studies, we thus aim to extend our analysis to more cyclones to understand their driving processes but this time from the perspective of impacts."*

2. The diabatic heating, which was found second most important cyclogenetic factor here, is expected to depend on the SST, which has a seasonal distinct course. I suggest you to inquire the dependence of this factor on the time of the year.

This is a pertinent comment (please also refer to third Reviewer's comment about seasonal dependence of PV profiles). In Fig. R1, we show a scatterplot that relates SST close to the centres of the 100 cyclones to the latent heat-produced PV at 850 hPa at the time of their mature stage. No significant correlation is found, although higher SSTs seem to be indeed related to higher values of $q_{cu}+q_{mp}$. As explained right below, it is also rather difficult to use Fig. R1 to reach to conclusions about the seasonal dependence of $q_{cu}$ and $q_{mp}$ and SST, despite the seemingly distinct clusters of Spring and Autumn cyclones in lower and higher SSTs, respectively. Taking into consideration the important role of SST in cyclones intensity, we added the following discussion:

*"Finally, despite the important role of sea surface temperature (SST) in modulating convection (e.g. Miglietta et al. 2011; Pytharoulis, 2018), in this study we found no seasonal dependence or direct relationship between SSTs and $q_{mp}+q_{cu}$ in low atmospheric levels (not shown). This is partly due to the diversity of the 100 cyclones' dynamics and partly due to the fact that all systems occur in different regions and different seasons. For instance, convection might differ between two systems with similar underlying SSTs due to different large scale forced ascent. Similarly, two cyclones that occur in the same season, but in different areas are very likely to present considerable differences in their underlying SSTs and consequently to convection. In fact, the investigation of the role of SST in cyclone dynamics from the perspective of PV tracers is not a straightforward issue due to the trait of $q_{mp}+q_{cu}$ PV tracers to accumulate the diabatically produced PV. As a result, an air mass that has an impact on cyclone dynamics might have experienced the PV gains due to latent heat in remote areas to cyclones centre."*

[Figure]

**Fig. R1 Scatterplot relating the sum of $q_{cu}$ and $q_{mp}$ for all 100 cyclones with SST, averaged within an area of 900 km$^2$ around the cyclone's centre. Values of the PV tracers correspond to averages within a 100 km radius around the cyclone centres at the 850 hPa level. The size of the dots is proportional to the mean sea level pressure at the cyclone's centre and dots are coloured according to the season of cyclones' occurrence: black for Winter, magenta for Spring, green for Summer and blue for Autumn.**

Miglietta, M. M., Moscatello, A., Conte, D., Mannarini, G., Lacorata, G. and Rotunno, R.: Numerical analysis of a Mediterranean 'hurricane' over south-eastern Italy: Sensitivity experiments to sea surface temperature, Atmospheric Research, 101(1–2), 412–426, https://doi.org/10.1016/j.atmosres.2011.04.006, 2011.

Pytharoulis, I.: Analysis of a Mediterranean tropical-like cyclone and its sensitivity to the sea surface temperatures, Atmospheric Research, 208, 167–179, https://doi.org/10.1016/j.atmosres.2017.08.009, 2018.

3. The baroclinic factor is influenced by the topography surrounding the Mediterranean. Hence you may find a geographical variability in the contribution of this factor in the various parts of the Mediterranean.

We agree with the Reviewer that the role of mountains in cyclones intensification is relevant to cyclones development, especially during cyclogenesis. This is relevant to the low-level PV streamer

of high $q_{co}$ values and the $q_{mo}$ PV-tracer. To enrich the discussion we included the following comment in section 5.3:

*"In fact, baroclinic forcing and topographic PV production contribute both to lee cyclogenesis. However, disentangling their relative contributions is a rather challenging issue since both processes take place in parallel (Buzzi et al. 2020 and references therein). Nevertheless, PV tracers may provide further insights into this complex problem. As discussed in section 4.1, mountains are likely to play the role of a "constant" PV source due to their interaction with the air flow. This constant PV source is imprinted into the simulations' initial conditions and corresponds to the low-level streamer of $q_{co}$ that wraps cyclonically around the cyclone centres in Fig. 5b. However, Fig. 15 shows that large part of high PV values close to mountains is related to $q_{mo}$ and thus to Ekman and baroclinic PV production (last two terms in Eq. 8). In these regards, PV production plays an important role in lee cyclogenesis. This is consistent with the sensitivity tests of McTaggart-Cowan et al. (2010b) who showed that the removal of the Alps inhibited cyclogenesis of a medicane.*

*After the stage of cyclogenesis, Mediterranean cyclones are still influenced by mountains interactions with the air flow due to the complex geography of the region and its sharp land-sea transitions. For instance, an air mass might reinforce a cyclone when it reaches to its centre due to experiencing earlier PV production while interacting with topography. However, the time period during which air masses sustain their high values of $q_{mo}$ is rather uncertain, i.e. the "dynamical memory" of $q_{mo}$, similar to that discussed for latent heating in section 4.2. Therefore, it is rather difficult to discern the areas where the influence of mountains is more important. Such analysis would benefit from a Langrangian approach (e.g. Attinger et al. 2019) where the evolution and origin of high values of $q_{mo}$ could be backtracked to mountains. However, this is currently out of the scope of this study and will be the subject of a following research study."*

**Reply to anonymous Reviewer #3**

*This study presents an in-depth and thorough analysis of Mediterranean cyclones from a potential vorticity (PV) perspective. The authors use a very rigorous approach by computing an on-line PV budget in the Weather, Research, and Forecasting (WRF) model, along the lines of Stoelinga (1996). The study presents a large number and variety of cases derived from an ERA5 climatology, and simulated with WRF to collect PV information associated with various physical processes (including physics parameterizations). Then, PV inversion was used to quantify the contributions of different physical processes to the cyclone's circulation. Results demonstrate the varying contributions of different processes, including "Medicane" systems that are shown to rely heavily, but not always exclusively on diabatic processes. The conclusions and interpretations in this study are strongly supported by rigorous analysis and clear graphics. In sum, I review many papers, and this is one of the stronger papers I've reviewed recently. I enjoyed reading it.*

*However, there are ways that the study could be strengthened and clarified, though I view these suggestions as minor revisions. In terms of the WCD review criteria, I would rate the Scientific significance as "good" (see main suggestion #1 below), Scientific quality as "excellent", and Presentation quality is good to excellent.*

We are thankful for the positive review and for carefully reading our manuscript. We found the comments to be very constructive and helpful to improve our article.

*Main suggestions:*

*1) While the goal and focus of the study is on Mediterranean cyclones, the results seem disconnected from other studies that have used PV-based methods to quantify cyclone dynamics in different geographical regions. I was surprised that even some of the PV-based cyclone classification studies that include the second author were not cited, for example (e.g. Gray and Dacre 2006). Also, none of Ahmadi-Givi et al (2004), Deveson et al. (2002), Plant et al. (2003) were cited. Placing your Mediterranean results within the context of existing literature using similar methods in different would strength your conclusion sections, where you allude to such connections. This would elevate the scientific significance of your study.*

To increase the link of our study with the efforts of the broader community working on cyclones, we have further developed the last paragraph of the introduction. It now reads:

*"Cyclones may be conceptualized as the outcome of PV anomalies that extend to different atmospheric levels. As a diagnostic variable, PV has been especially helpful to provide deep insights into the development mechanisms of different types of cyclones, including tropical cyclones (e.g. Möller and Montgomery, 2000; Chan et al. 2002) and mid-latitude storms (e.g. Boettcher and Wernli, 2011). As a result, the quantification of the relative contribution of different atmospheric processes to the total atmospheric PV is equivalent to delineating the contribution of these processes to cyclone dynamics. For instance, Gray (2006), Chagnon and Gray (2015) and more recently Spreitzer et al. (2019) used PV budget diagnostics to analyse the impact of diabatic processes to the dynamical structure of the upper troposphere. In the same direction, several studies used PV-budget diagnostics to investigate the role of convection and more generally of latent heat in the dynamical structure of cyclones (Martínez-Alvarado et al. 2016; Büeler and Pfahl, 2017; Attinger et al. 2019). By extension, piecewise PV inversion of these individual contributions is expected to reproduce the partition of the low-level wind circulation that is related to each atmospheric process, i.e. the contribution of these processes to the cyclone itself (defined as a vortex).*

*Applying piecewise PV inversion to individual PV tracers is a well-established approach to quantify the relative contribution of different processes to the development of cyclones (e.g. Davis and Emanuel 1991; Wu and Emanuel 1995; Stoelinga, 1996; Huo et al. 1999; Bracegirdle and Gray 2009; Schlemmer et al. 2010; Seiler, 2019). For instance, Stoelinga (1996) and Ahmadi-Givi et al. (2004) quantified the contribution of diabatic processes to the intensification of two mid-latitude storms. The use of PV-budget diagnostics or piecewise PV inversion to perform a process-based classification of cyclones is still missing from the state-of-the-art. Such an approach would extend the analysis of Čampa and Wernli (2012) who analysed the vertical PV profile of extratropical cyclones of different geographical origin and would be complementary to earlier efforts by Deveson et al. (2002), followed by Plant et al. (2003) and Gray and Dacre (2006) where extratropical cyclones were classified to three types according to the contribution of upper and lower troposphere to their development. Despite the considerable advancements in analysing and classifying tropical and mid-latitude cyclones from a PV perspective, Mediterranean cyclones have been rarely been the object of comprehensive studies in which their dynamical structure is analysed in a climatological context. In this study, we use regional climate modelling to simulate 100 of the most intense Mediterranean cyclones and we analyse their dynamics using PV-budget diagnostics and piecewise PV inversion."*

*2) The process you call "momentum forcing" is unclear, and possibly incomplete. What, exactly, is this process? Can you please explain this more clearly, perhaps with a schematic? Also, you selected a convective parameterization scheme that doesn't adjust momentum (Kain-Fritsch), and so convective momentum adjustment is missing in your model runs. Perhaps the slack is picked up by the turbulence parameterization? It would be interesting to see if there would be differences in the online PV budget if you used one of the WRF schemes that also adjusts momentum (e.g., Zhang-McFarlane, Arakawa-Schubert, Tiedtke...)*

Thank you for this comment, we agree that such a new mechanism for cyclones' intensification deserves more elaboration. We performed important changes in section 4.2 to further support the reader with information about the role of $q_{mo}$ in reinforcing cyclonic circulation. In addition, we performed an additional simulation for case #62 (one of the two peculiar cases in section 5.4), where instead of Kain-Fritsch, we used the Tiedtke cumulus parametrisation and thus we introduced an additional PV tracer related to the momentum adjustments applied to the wind field due to convection. The new PV tracer ($q_{mocu}$) was related to rather weak values of PV and $q_{mo}$ was still the one presenting the highest values. For reasons of comparison, right below, Fig. R2 shows $q_{mo}$ and $q_{mocu}$ at 850 hPa and at the time of the cyclone mature stage. Further analysis showed that values of $q_{mocu}$ never exceeded 0.2 PVUs in absolute values, close to the centre of the cyclone.

[Figure]

**Figure R2 fields of wind, $q_{mo}$ and $q_{mocu}$ at 850 hPa at the time of cyclone #62 mature stage. Please note that colorbar ranges are different in the two panels.**

Results from the new simulation are now presented in section 5.4 to provide further insights into the areas of generation of positive values of $q_{mo}$. Here follows the revised part of section 4.2, dedicated to $q_{mo}$, and the additional information for the peculiar case #62 in section 5.4

**Section 4.2:**

[revised manuscript text omitted]

**Section 5.4**

*3) There are some other classic Alpine lee cyclone studies of Mediterranean cyclones that are not cited, for example, especially Bleck and Mattocks (1984), who used a PV-based approach to study this phenomenon:*

*Bleck, R., & Mattocks, C. (1984). A preliminary analysis of the role of potential vorticity in Alpine lee cyclogenesis. Beiträge zur Physik der Atmosphäre, 57(3), 357-368.*

Thank you for this suggestion. Bleck and Mattocks is now cited and along with other changes as suggested below, the introduction now reads:

*"On the other hand, the high mountains that surround the Mediterranean basin have a direct effect on cyclones by interacting with the air flow and triggering PV streamers of topographic origin.*

*Such streamers are usually referred to as PV banners (e.g. Aebischer and Schär,1998) and correspond to filaments of diabatically produced PV at the wake of mountains (Rotunno et al. 1999; Epifanio and Durran, 2002; Schar et al. 2003; Flamant et al. 2004). The potentially important role of such PV production in Alpine cyclogenesis, one of the most cyclogenetic areas of the Mediterranean basin, has been long ago suggested to be important (e.g. Bleck and Mattocks, 1984) and have been recently reviewed by Buzzi et al. (2020). Especially when combined with the upper-tropospheric PV streamer, PV banners may even reinforce cyclones (Tsidulko and Alpert, 2001; Mc-Taggart-Cowan et al. 2010a, 2010b). Therefore, the uniqueness of the region adds a higher degree of complexity to the processes implicated in Mediterranean cyclone development."*

**Minor corrections and suggestions:**

*Line 40: "Topographic PV streamers" comes to my mind, but they are "banners" here. Are the terms used here consistent with previous literature? E.g., Aebischer and Schär (1998).*

It seems that there is a typo and the Reviewer probably refers to line 72 of the introduction in the discussions paper where we first mention the word "banner".

A high number of articles refer to "banners" even in their titles. Here follow two recent examples:

Siedersleben, Simon K., and Alexander Gohm. "The missing link between terrain-induced potential vorticity banners and banded convection." *Monthly Weather Review* 144.11 (2016): 4063-4080.

Bader, R., Sprenger, M., Ban, N., Rüdisühli, S., Schär, C., & Günther, T. (2019). Extraction and Visual Analysis of Potential Vorticity Banners around the Alps. *IEEE Transactions on Visualization and Computer Graphics, 26*(1), 259-269.

We agree though that the term should be more properly introduced for readers who are not familiar with it. We rephrased as follows:

*"On the other hand, the high mountains that surround the Mediterranean basin have a direct effect on cyclones by interacting with the air flow and triggering PV streamers of topographic origin. Such streamers are usually referred to as PV banners (e.g. Aebischer and Schär,1998) and correspond to filaments of diabatically produced PV at the wake of mountains (Rotunno et al. 1999; Epifanio and Durran, 2002; Schar et al. 2003; Flamant et al. 2004)."*

*Line 49: Suggest adding modifier "Mediterranean" before "cyclones"*

Done.

*Line 67: It may be a minor, semantic point, but I would disagree that convection affects all cyclones, unless you are including slantwise convection perhaps. Upright convection is often, but not always present. Some cyclones, such as "heat lows" in the South-western US, can be accompanied by clear skies, albeit perhaps with dry convection during daytime hours. Some lee cyclones may be devoid of convection though.*

Agreed, our phrase made indeed a very strong statement. We rephrased as follows:

*"Baroclinic instability and convection are two of the most common processes that affect the dynamics of extratropical cyclones."*

*Lines 110-112: I am curious how you implemented the on-line PV code with the WRF hybrid coordinate. The PV tendency equations would then have to vary with the vertical coordinate. This must have been very difficult, technically, and I am impressed. I would love to work with this module, as would many others I'm sure – is the code available to share?*

Thank you for this comment. We would be glad to share the code with colleagues and collaborate for its further development. According to WCD manuscript preparation rules, we included the following section after the "Acknowledgements" part:

*"**Code availability:** The PV-tracers diagnostic corresponds to a single module, written in Fortran and its implementation requires additional modifications to several parts of the original WRF code. The whole PV-tracers package is adapted to WRF version 4.0 and might demand minor, additional modifications for its implementation in subsequent versions of the model. The diagnostic and all relevant source code modifications are freely available upon request."*

*Lines 113-115: Again, somewhat minor, but I would not recommend simulating tropical-like systems with a 5-class microphysics scheme. Without hail or graupel, snow is unrealistically lofted in strong, convectively forced systems with updraft speeds greater than a m/s or so.*

Thank you for this recommendation. We included the following:

*"This combination of physical parametrisation was shown to adequately reproduce Mediterranean cyclones (e.g. Flaounas et al., 2019; Fita and Flaounas, 2018; Miglietta and Rotunno, 2019). Especially the use of a five-class microphysics scheme was shown to yield simulations with a small track error in two medicane cases, when compared to the use of other microphysical parametrisations (Miglietta et al, 2015; Pytharoulis et al., 2018). It is however noteworthy that the absence of hail or graupel results to unrealistic representation of the microphysical properties of deep convective clouds."*

*Section 2.2, around line 170: Citation of some prior work may be useful here, such as the Schubert et al. 2004 paper: Schubert, W.H., Hausman, S.A., Garcia, M., Ooyama, K.V. and Kuo, H.C., 2001. Potential vorticity in a moist atmosphere. Journal of the atmospheric sciences, 58(21), pp.3148-3157.*

Thank you for this comment. In the end of Section 2.2 we included the following:

*"Finally, it is noteworthy that version 4.0 of the WRF model uses moist potential temperature to allow consistent treatment of moisture in the calculation of pressure in its dynamical core. For the purposes of our analysis we could also use the moist potential temperature as a basis for calculating a generalized PV field for a moist atmosphere. Schubert et al., (2001) showed that such a moist PV field still complies with the invertibility principle and thus it incorporates all information that defines the atmospheric state. However, to be consistent with previous studies, we performed our analysis with all PV fields in WRF being calculated with potential temperature."*

*Line 183: Check spelling on Stoelinga*

Done.

*Line 189: The 3-day time average is quite a lot shorter than Davis' 7-day average. No need to re-do anything, but what are the implications? My thought is that some upper-level PV contributions are lost, as the anomaly computed as a deviation from a 3-day average will be weaker for large, strong, persistent upper troughs.*

Also in accordance with the comment of Reviewer #1, we added the following:

*"A three-day time period might be short to define PV anomalies in cases of slowly evolving upper tropospheric systems. However, anomalies need to be consistently defined for all 100 cyclones and Mediterranean cyclones are systems of relatively short lifespan. Therefore, we consider that a three-day average is a fair compromise for the application of piecewise PV inversion in this study."*

*Figure 1: Note typo in caption "smean ea"*

Apologies for this typo. It is now corrected.

*Line 240: The position of cold-frontal, lower-tropospheric cyclonic PV maxima will vary from case to case, and so in a composite, there will be a large degree of cancellation (e.g., the type of feature highlighted by Lackmann 2002): Lackmann, G.M., 2002. Cold-frontal potential vorticity maxima, the low-level jet, and moisture transport in extra-tropical cyclones. Monthly Weather Review, 130(1), pp.59-74. This is also mentioned and discussed near lines 325-327.*

We agree with the Reviewer that there will be a certain degree of cancellation due to the composite averaging. However, frontal gradients of $\theta e$ are still evident in our Figs 3 and 7 and they are also similar to the ones of extratropical cyclones (Dacre et al. 2012). In addition, they are consistent with the ones in Flaounas et al. (2015) where Mediterranean cyclone fields were rotated before performing composite averages to present consistent frontal structures between them. We added the following in section 4.2 to enrich the discussion:

*"It is noteworthy that although frontal structures are evident in Figs 3 and 7, their exact location, intensity and vertical PV profile are case dependent. As a result, latent heating-produced PV maxima along the cold fronts and their associated PV tendencies right below and above these maxima (Lackmann 2002) will be at some extent inconsistent between the cyclones. This will affect values in the composite averages of Fig. 7a. Such inconsistencies are also expected in composite averages of other PV tracers close to the surface (e.g. Figs 7c and 7e), where PV is typically negative in the cold sector, due to small or negative static stability, and positive values in the warm sector (Vannière et al. 2016)."*

Dacre, H. F., Hawcroft, M. K., Stringer, M. A. and Hodges, K. I.: An extratropical cyclone atlas: A tool for illustrating cyclone structure and evolution characteristics, Bulletin of the American Meteorological Society, 93(10), 1497–1502, https://doi.org/10.1175/BAMS-D-11-00164.1, 2012.

Flaounas, E., Raveh-Rubin, S., Wernli, H., Drobinski, P. and Bastin, S.: The dynamical structure of intense Mediterranean cyclones, Clim. Dyn., 44(9–10), 2411–2427, doi:10.1007/s00382-014-2330-2, 2015.

Lackmann, G. M.: Cold-Frontal Potential Vorticity Maxima, the Low-Level Jet, and Moisture Transport in Extratropical Cyclones, Mon. Weather Rev., 130, 59–74, https://doi.org/10.1175/1520-0493(2002)130<0059:cfpvmt>2.0.co;2, 2002.

Vannière, B., Czaja, A., Dacre, H., Woollings, T. and Parfitt, R.: A potential vorticity signature for the cold sector of winter extratropical cyclones, Q.J.R. Meteorol. Soc., 142(694), 432–442, https://doi.org/10.1002/qj.2662, 2016.

*Line 265: At some point, it seems that diurnal changes in PBL depth, and the fact that PV ~ 0 in the PBL should be mentioned or discussed. Or perhaps better near line 360.*

Thank you for this comment. We included the following in section 4.2:

*"Indeed, Fig. 6b shows a positive vertical gradient in the average profile of temperature turbulent fluxes ($T_{bl}$) in the first model levels. However, it is noteworthy, that the vertical profile of $T_{bl}$ is subject to diurnal variabilities of the PBL height affecting thus its vertical gradient and the magnitude of qbl. In addition, $q_{bl}$ presents a high variability in amplitude and sign over the cyclone centres in Fig. 4f."*

*Line 285: I wonder if there is any seasonality to the varying contributions?*

We repeated Fig. 4 showing now the average of all 100 cases according to their season of occurrence. This version of Fig. 4 is shown below (Fig. R2). It seems from Fig. R2 that some superficial conclusions could be drawn. For instance, from Autumn to Summer, $q_{lw}$ in the first model levels are gradually displaced from positive to negative values and vice versa for $q_{bl}$. Such a displacement is also consistent with the latent heat PV tracer profiles of $q_{mp}+q_{cu}$. Despite these patterns, it is quite delicate to draw results from Fig. R2 regarding the seasonal dependence of Mediterranean cyclone dynamics. First, this is due to the small size of the cyclone population that limits the performance of any meaningful statistics (e.g. only three summer cyclones are included in our 100 cases). Furthermore, cyclones are located in different locations (Fig. 1) and thus our analysis will be influenced by the high spatial variability of SSTs in the region (please also refer to the relevant query of the second Reviewer). Finally, the mature stage of several of our 100 cyclones takes place over land, or close to mountain areas, influencing accordingly their thermodynamics e.g. latent heat release in forced ascent. For these reasons we consider that the variety of locations and subcategories of Mediterranean cyclones in our 100 cases renders the comparison of cyclone dynamics rather unfair and thus we chose not to include Fig. R2 in our analysis.

[Figure]

**Figure R2 Vertical profiles of total PV and PV tracers for all 100 cyclones (in grey lines), averaged 100 km around cyclone centres at the time of their mature stage. Profile averages and zero PVU are shown in solid and dashed lines, respectively. Averages are shown according to the season of cyclones' occurrence: black for Winter, magenta for Spring, green for Summer and blue for Autumn. Note that range of values in x-axis is not common for all panels.**

*Line 458: Capitalize Gulf*

Done.

*Line 484: Is it Fig. 16c that should be referenced?*

Indeed, it is now corrected.

*Line 506: Can references be provided to support this sentence?*

The sentence now reads:

"*Medicanes have been long before proposed to share similar dynamics with tropical cyclones (e.g Emanuel, 2005; Fita et al., 2007).*"